# FTO-mediated cytoplasmic m$^6$A$_m$ demethylation adjusts stem-like properties in colorectal cancer cell

Sébastien Relier[1], Julie Ripoll[2], Hélène Guillorit[1,3], Amandine Amalric[1], Cyrinne Achour [4,5], Florence Boissière [6], Jérôme Vialaret[7,8], Aurore Attina[7,8], Françoise Debart [9], Armelle Choquet[1], Françoise Macari[1], Virginie Marchand [10], Yuri Motorin[10], Emmanuelle Samalin[1,6], Jean-Jacques Vasseur[9], Julie Pannequin[1], Francesca Aguilo[4,5], Evelyne Lopez-Crapez[6], Christophe Hirtz [7,8], Eric Rivals [2,11✉], Amandine Bastide [1,11✉] & Alexandre David [1,7,11✉]

Cancer stem cells (CSCs) are a small but critical cell population for cancer biology since they display inherent resistance to standard therapies and give rise to metastases. Despite accruing evidence establishing a link between deregulation of epitranscriptome-related players and tumorigenic process, the role of messenger RNA (mRNA) modifications in the regulation of CSC properties remains poorly understood. Here, we show that the cytoplasmic pool of fat mass and obesity-associated protein (FTO) impedes CSC abilities in colorectal cancer through its N$^6$,2'-O-dimethyladenosine (m$^6$A$_m$) demethylase activity. While m$^6$A$_m$ is strategically located next to the m$^7$G-mRNA cap, its biological function is not well understood and has not been addressed in cancer. Low FTO expression in patient-derived cell lines elevates m$^6$A$_m$ level in mRNA which results in enhanced in vivo tumorigenicity and chemoresistance. Inhibition of the nuclear m$^6$A$_m$ methyltransferase, PCIF1/CAPAM, fully reverses this phenotype, stressing the role of m$^6$A$_m$ modification in stem-like properties acquisition. FTO-mediated regulation of m$^6$A$_m$ marking constitutes a reversible pathway controlling CSC abilities. Altogether, our findings bring to light the first biological function of the m$^6$A$_m$ modification and its potential adverse consequences for colorectal cancer management.

[1] IGF, Univ. Montpellier, CNRS, INSERM, Montpellier, France. [2] LIRMM, Univ. Montpellier, CNRS, Montpellier, France. [3] Stellate Therapeutics, Paris, France. [4] Wallenberg Centre for Molecular Medicine (WCMM), Umea University, Umea, Sweden. [5] Department of Medical Biosciences, Umea University, Umea, Sweden. [6] ICM, Montpellier, France. [7] IRMB-PPC, Univ. Montpellier, INSERM, CHU Montpellier, CNRS, Montpellier, France. [8] INM, Univ. Montpellier, INSERM, Montpellier, France. [9] IBMM, CNRS, Univ. Montpellier, ENSCM, Montpellier, France. [10] Université de Lorraine, IMoPA UMR7365 CNRS-UL and UMS2008/US40 IBSLor, UL-CNRS-INSERM, BioPole, Vandoeuvre-les-Nancy, France. [11] These authors jointly supervised this work: Eric Rivals, Amandine Bastide, Alexandre David. ✉email: rivals@lirmm.fr; amandine.bastide@igf.cnrs.fr; alexandre.david@igf.cnrs.fr

Despite significant advances in diagnosis and therapy, colorectal cancer (CRC) remains a major cause of mortality and morbidity worldwide. CRC survival is highly dependent upon early diagnosis. Patients with localized cancer exhibit 70–90% 5-year survival. Survival from metastatic cancer plummets to 10%. Metastasis is a multistep process encompassing local infiltration of tumor cells into adjacent tissues, transendothelial migration into vessels, survival in the circulatory system, extravasation, and colonization of secondary organs[1]. This process entails constant reprogramming of gene expression to enable tumor adaptation in different environments, a peculiar trait of cancer stem cells (CSCs). CSCs constitute a minor subpopulation of tumor cells endowed with self-renewal and multilineage differentiation capacity[2]. The most clinically relevant trait of CSCs is their ability to metastasize and escape from standard chemotherapy[3]. Understanding the molecular mechanisms that participate to the CSC phenotype is critical to designing improved cancer therapeutics.

Among more than 100 post-transcriptional modifications reported to occur on RNA[4], N6-methyladenosine ($m^6A$) is the most frequent modification of mammalian messenger RNAs (mRNAs)[5]. $m^6A$ is involved in all post-transcriptional steps of gene expression (mRNA splicing, transport, stability, and translation) and plays a role in pleiotropic biological processes including development, immunology, and stem cell biology[5]. Therefore, it comes at no surprise that $m^6A$ dysregulation is intricately involved in the progression of several solid and non-solid tumors while the underlying functional mechanism differs from one malignancy to another[6–11].

Discovered several decades ago[12–16], the function of $m^6A$ remained obscure until the identification of the first $m^6A$ demethylase, the fat mass and obesity-associated protein (FTO)[17]. The marriage of immunochemical approaches with next-generation sequencing (NGS) technologies revealed the unique topology of $m^6A$ distribution along mRNA. $m^6A$ is a dynamic reversible chemical modification catalyzed by a protein complex consisting of the methyltransferase-like 3 and 14 (METTL3 and METTL14), and several auxiliary proteins such as the Wilms' tumor 1-associating protein (WTAP)[18–21]. Effects of $m^6A$ involve recruitment of reader proteins, e.g., YTHDF1[22] or YTHDF2[23], that lodge the modified adenosine in the hydrophobic pocket of their YTH domain. Finally, $m^6A$ is removed by the AlkB homolog 5 (ALKBH5)[24] and FTO[17] (Fig. 1a). $m^6A$ modification generally occurs in a subset of RRA*CH consensus sites (R, purine; A*, methylable A; C, cytosine; H, non-guanine base), at the beginning of the 3′-UTR near the translation termination codon[25,26] with one exception. Indeed, besides internal adenosine, 2′-O-methyladenosine ($A_m$) residue adjacent to the N7-methylguanosine ($m^7G$) cap, can be further methylated at the N6 position and become the N6,2′-O-dimethyladenosine ($m^6A_m$)[27]. $m^6A_m$ can be deposited by the recently identified PCIF1/CAPAM[28] and removed by FTO[29]. Unlike $m^6A$, the biological function of $m^6A_m$ mRNA modification is poorly understood and its potential involvement in cancer onset or evolution has never been addressed.

Here, we uncover the importance of cytoplasmic FTO-mediated $m^6A_m$ level adjustment to colorectal CSC. Using patient-derived cell lines, we establish that FTO activity hampers CSC abilities through an unconventional process that does not seem to trigger significant reprogramming of basal gene expression.

## Results

**FTO inhibition promotes stem-like traits in colorectal cancer cell lines.** We initially evaluated the involvement of $m^6A$ modification in generating the CSC phenotype. Due to their inherent plasticity, CSCs are best identified via their functional abilities, such as tumorigenic potential and chemoresistance, rather than surface biomarkers. We examined the ability of short interfering RNAs (siRNA) targeting known $m^6A$ mediators—writers, readers and erasers—to alter sphere-forming potential (SFP) (Fig. 1b). SFP is the ability of cancer cells—from either conventional cell lines or patient exeresis—to form microtumor-like spheroids (colonospheres) from a single cancer progenitor cell[30]. This model is often used as a surrogate to evaluate the tumorigenic potential of solid tumors (5). We used CRC1 cells, a colorectal cancer cell line established in the lab from colorectal cancer exeresis[31]. The various siRNAs significantly silenced expression of individual target genes: METTL3, METTL14, WTAP, YTHDF1, YTHDF2, ALKBH5, and FTO (Fig. S1a). Only FTO knockdown affected SFP, nearly doubling colonospheres numbers (Fig. 1c). We confirmed the importance of FTO using a different targeting siRNA (Fig. 1d and Fig. S1b) and three other cell lines, derived from primary (HCT-116) and metastatic (CPP19 and SW620) tumors (Fig. 1d).

Next, we generated stable CRC1 cell lines expressing GFP with either a short hairpin RNA targeting FTO (sh-FTO) or an irrelevant scrambled short hairpin control (sh-CTL), and selected transfected clones by cell sorting for GFP expression (Fig. S1c). Sh-FTO expressing cell lines established from individual cells exhibited a three to ten-fold decrease of FTO expression as shown by immunoblot analysis (Fig. S1d) and displayed increased SFP, confirming the siRNA findings (Fig. S1e). Importantly, FTO knockdown does not influence cell growth (Fig. S1f), which differs from the AML (acute myeloid leukemia) model[8]. Then, we evaluated the level of several stemness-related markers associated with CSC features. FTO knockdown cells displayed enhanced aldehyde dehydrogenase (ALDH) activity, a hallmark of CSC[32] (Fig. 1e), and increased expression of CD44 and CD44v6 (Fig. 1e), membrane receptors associated with tumor progression[33] and indispensable for CSC tumor initiation, chemoresistance, and epithelial to mesenchymal transition. CRC1 sh-FTO cells demonstrated a ten-fold increase in the number of CD44+ cells (from 2.1 to 21%) (Fig. 1e) and CD44v6 isoform expression was doubled (Fig. 1e). FTO knockdown in SW620 cells phenocopied the CRC1 cells (Fig. S1c–h).

To connect FTO levels with in vivo tumor initiation potential we inoculated immunodeficient mice (athymic nude) with increasing numbers of sh-FTO or sh-CTL cells (Fig. 1f). Seven weeks later, tumor xenografts (diameter > 100 $mm^3$) were counted and harvested. Remarkably, as few as hundred sh-FTO cells were capable of initiating tumor formation in 4/5 mice vs. 1/5 for sh-CTL cells (Fig. 1g). Extreme limiting dilution software analysis (ELDA) shows that the frequency of tumor-initiating cells is seven-fold higher in sh-FTO cells, with a highly significant $p$-value ($p = 0.00035$) (Fig. 1g).

Based on these findings, we conclude that diminished FTO expression promotes the CSC phenotype.

**FTO silencing confers resistance to chemotherapy in colorectal cancer cell lines.** We extended these findings to chemoresistance, another hallmark of CSC. We employed FIRI treatment, a combination of 5-fluorouracil and irinotecan, used to treat metastatic colorectal cancer[34]. We followed a standard protocol for treating colorectal cancer cell lines based on 3 days treatment with FIRI (50 μM 5-flurouracil (5-FU) + 500 nM SN38, active metabolite of irinotecan). siRNA mediated FTO targeting conferred significant chemoprotective effects on CRC1 cells (2 to 3-fold) in comparison with control cells (si-CTL)(Fig. 2a, left). As above, we obtained a similar effect in SW620 (Fig. 2a, right), as well as in stable sh-FTO models (Fig. 2b).

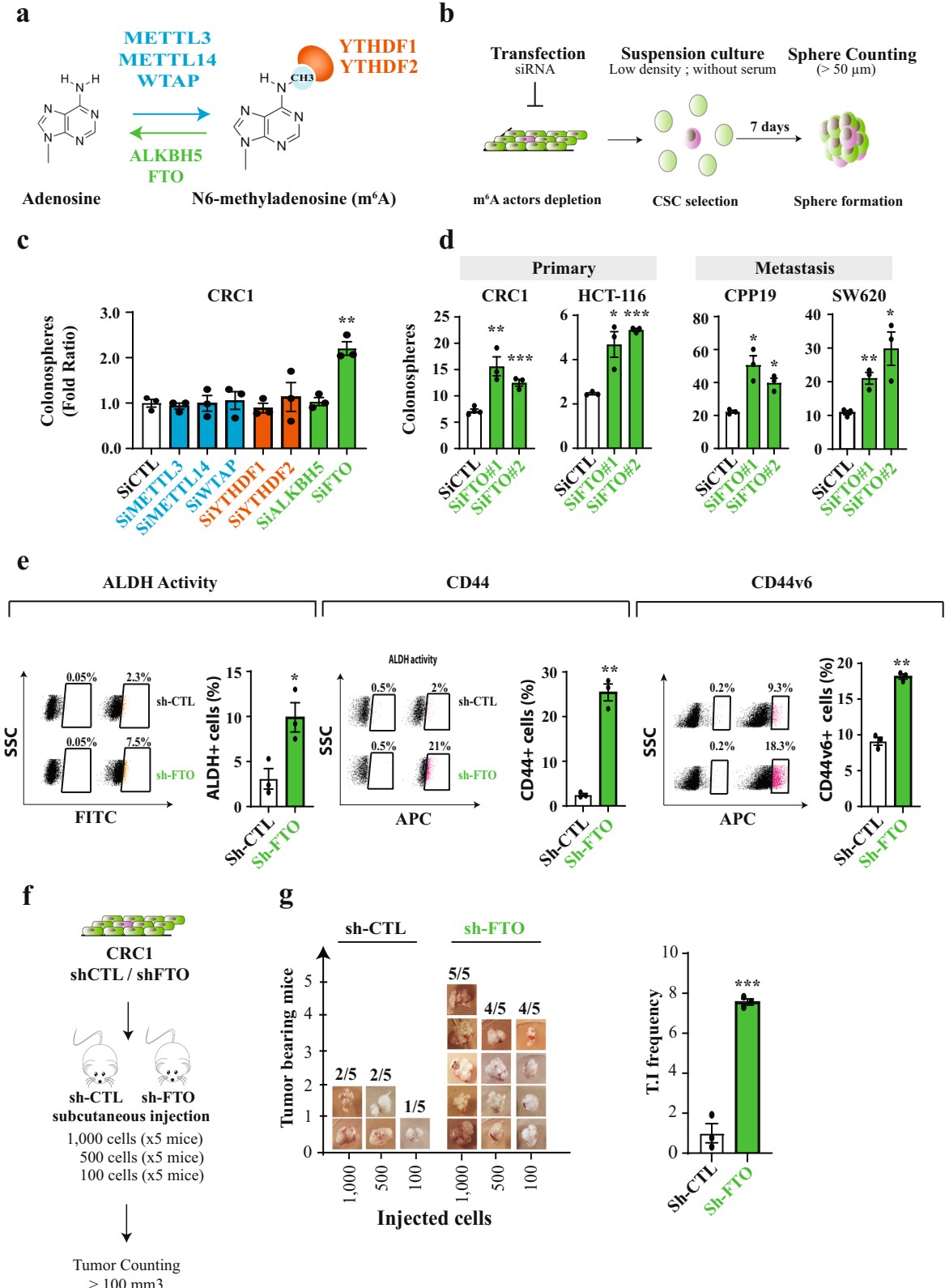

Importantly, chemoprotection bestowed by FTO silencing extended to FOX treatment (50 μM 5-flurouracil (5-FU) + 1 μM oxaliplatine), another advanced stage colorectal cancer therapy[35,36] (Fig. S2a). To evaluate the effect of FTO knockdown in vivo we injected sh-CTL or sh-FTO cells (50,000 cells) into nude mice (six mice per group). When the tumors reached about 100 mm³ (day 21), we treated mice with FIRI (50 mg/kg 5-FU + 30 mg/kg

irinotecan) and measured tumors twice a week for four and a half weeks (Fig. 2c, left). While FIRI treatment stabilized tumor size in sh-CTL mice, sh-FTO tumors displayed substantial chemoresistance maintaining consistent growth (Fig. 2c, right).

Do chemoresistant cells display low FTO levels? To produce CSC-enriched chemoresistant cells, we treated CRC1 and SW620 cells for three days with sub-lethal doses of FIRI (0.2× FIRI =

**Fig. 1 FTO inhibition promotes stem-like traits in colorectal cancer cell lines. a** Actors of m6A modification targeted by the siRNA screening. The writer complex METTL3—METTL14—WTAP deposits m6A while ALKBH5 and FTO erases m6A. Both YTHDF1 and YTHDF2 are readers of m6A modification. **b** Sphere forming assay. siRNA-transfected cells were seeded at low density (1 cell/μL) in non-adherent conditions and serum-deprived medium. This type of suspension culture allows the survival and growth of stem-like/progenitor cells. Following 7 days of culture, the number of spheres correlates with the initial number of CSC. **c** FTO silencing increases the sphere forming potential of CRC1 cell line. Colonosphere formation was quantified following knockdown of the main m6A actors. Results are expressed in fold change compared to si-CTL. $n = 3$ biological replicates. Mean ± S.E.M, **$p$-value < 0.01. Two-sided unpaired $T$-test. **d** FTO silencing increases the sphere forming potential in four CRC cell lines. Colonospheres quantification after silencing of FTO by two distinct siRNA, in four cell lines derived from primary tumors or metastasis. Results are expressed in fold change compared to si-CTL ($n = 3$ biological replicates). Mean ± S.E.M, *$p$-value < 0.05; **$p$-value < 0.01. Two-sided unpaired $T$-test. **e** Expression of CSC biomarkers. Level of ALDH activity as well as cell-surface expression of CD44 and CD44v6 were evaluated by flow cytometry in CRC1-sh-FTO cell line vs. sh-CTL. Graphs show one representative biological replicate ($n = 3$). Bar plots show quantification of the number of ALDH, CD44, and CD44v6 positive cells from three biological replicates. Mean ± S.E.M, *$p$-value < 0.05, **$p$-value < 0.01. Two-sided unpaired $T$-test. **f** In vivo tumor initiation assay. Either 1000, 500, or 100 CRC1 sh-FTO/sh-CTL cells were subcutaneously injected into nude mice. After 7 weeks, the number of tumor-bearing mice (tumor > 100 mm$^3$) was evaluated. **g** FTO silencing increases tumor initiation in vivo. Picture represents the number of tumor bearing mice for each group (five mice per group) after injection of sh-FTO or sh-CTL CRC1 cell line. Bar plot represents the quantification of tumor initiation (T.I) frequency obtained by ELDA software. ***$p$-value < 0.001, two-sided unpaired $T$-test.

10 μM 5-FU + 0.1 μM SN38). As expected, 0.2× FIRI treatment triggered cell cycle arrest at G2/M phase and increased subG1 cells (Fig. S2b). Surviving cells demonstrated increased ALDH activity (Fig. S2c) and enhanced resistance to chemotherapy (Fig. S2d). While variations of METTL3 and ALKBH5 levels could be noticed following chemoresistance acquisition, they were not consistent between cell lines (Fig. 2d). By contrast, FTO level was decreased by half in both cell lines (Fig. 2d). Noteworthy, the correlation between FTO mRNA expression levels and protein abundance varies from one cell line to another (Fig. S2e).

Together, these findings demonstrate that FTO expression is tightly linked to the CSC chemoresistant phenotype.

**FTO expression regulates CSC phenotype in circulating tumor cells.** Circulating tumor cells (CTCs) are often detected in the bloodstream of colorectal cancer patients, sometimes even at early stages of the disease[37]. CTC are responsible for metastasis and high mortality. We previously established CTC lines from chemotherapy-naïve stage IV (metastatic) CRC patients that display a strong CSC phenotype[31]. Immunoblotting revealed that FTO protein levels are reduced by ~50% in CTC lines in comparison with primary and metastatic patient derived lines (Fig. 3a). Decreased expression is achieved post transcriptionally, since quantitative PCR (qPCR) analysis of FTO mRNA levels showed no significant differences between CTC lines and primary/metastatic patient derived lines (Fig. 3b). A similar disconnect in FTO mRNA/protein was reported in gastric tumors[38].

Despite this, we could increase FTO protein by transfecting cells with a plasmid containing a FTO cDNA under a strong promoter. Increasing FTO in CTC lines (CTC44 and CTC45) did not affect cell proliferation in monolayer culture but decreased SFP (Fig. 3c), chemoresistance to FIRI treatment (Fig. 3d), and ALDH activity (Fig. 3e) in both cell lines.

Thus, multiple lines of evidence point to the conclusion that FTO is a key factor in maintaining the CSC phenotype.

**FTO functions via its m6A$_m$ demethylase activity.** How does FTO modulate CSC functions? First, we evaluated whether its catalytic activity was essential to achieve this phenotypic outcome. As FTO can demethylate m6A and m6A$_m$[29,39], we measured their levels in RNAse digested polyadenylated mRNA from both sh-CTL and sh-FTO cells using high-performance liquid chromatography-coupled to tandem mass spectrometry (LC-MS/MS) analysis. At first, we isolated mRNA from cell pellets and verified the purity of our sample preparation by LC-MS/MS (Fig. S3a) and qPCR (Fig. S3b). To detect m6A$_m$ proximal to

m7G-Cap, we used a previously published method based on including a decapping step (RNA 5′ Pyrophosphohydrolase, RppH treatment) prior to Nuclease P1 treatment[39]. We calibrated the assay using standard curves created with synthetic nucleoside standards (Fig. S3c). Noteworthy, we sometimes observed significant fluctuations of adenosine measurements, most probably because of the nature of the sample, enriched in poly-A tails. Further, adenosine signal from purified mRNA samples was very strong and could saturate the detector. We do not encounter this issue with other nucleosides (U, C, or G).

In both CRC1 and SW620 lines, FTO knockdown did not impact internal m6A/A level (Fig. 4a). However, we clearly observed a significant increase of m6A$_m$/A ratio (Fig. 4a). We performed controls to strengthen our observation: first, FTO silencing affected neither m6A/A nor m6A$_m$/A ratio in small RNA species, another potential target of FTO (Fig. S3d); second, to ensure our ability to detect m6A variation, we silenced METTL14, an m6A writer. As anticipated, METTL14 targeting triggered a significant decrease of m6A level by LC-MS/MS (Fig. S4a). Next, we performed a similar analysis with mRNA extracted from our panel of patients derived cell lines (Fig. 3b). We observed an increased level of m6A$_m$/A ratio in CTC lines (Fig. 4b) which was concomitant with low FTO levels (Fig. 3a). By contrast, m6A/A ratio was rather decreased (Fig. 4b). Importantly, FTO overexpression in two CTC lines triggered the opposite effect and decreased tremendously m6A$_m$/A ratio while m6A/A ratio remained unchanged (Fig. 4c). In order to ascertain whether FTO does not affect specific m6A sites, we employed methylated RNA immunoprecipitation sequencing (MeRIP-seq or m6A-seq)[25,26] from sh-CTL and sh-FTO cells. The enrichment of methylated RNA fragments was validated by RT-qPCR of target genes (Fig. S5a) as well as bioinformatics analysis (Fig. S5b–d, Table S7). As seen in Fig. S5d, enriched peaks were mainly found in CDS regions ~42%, followed by intronic and 3′-UTR regions (~27% and ~24%, respectively) with an enrichment near stop codons (Fig. S5e). However, comparison of the RNA methylome between the two cell lines did not reveal any significant change following FTO depletion, using $p$-value < 0.05 and log2 fold change of 1 (Table S8). Altogether, these observations established a first connecting thread between FTO-mediated m6A$_m$ dynamics and the acquisition of cancer stem ability in colorectal cancer.

Recent reports identified PCIF1/CAPAM as the m6A$_m$ methyltransferase ("writer")[28,40] (Fig. 4d) and proposed that PCIF1/CAPAM activity is involved in cellular resistance to oxidative stress response induction[28]. It is well known that elevated reactive oxygen species (ROS) production impairs self-renewal and promote cell differentiation in stem cells and their malignant counterpart[41]. To examine the contribution of PCIF1/

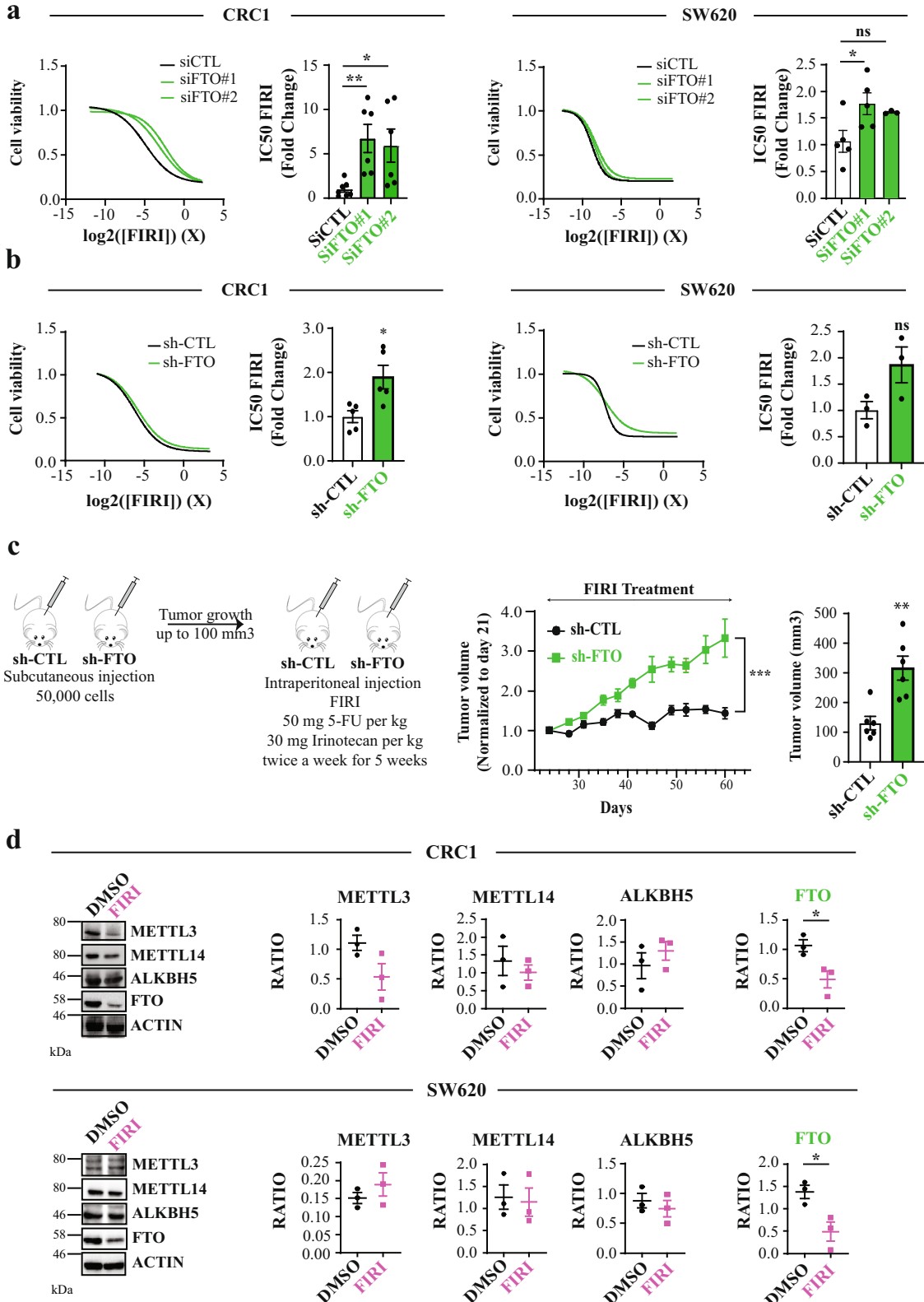

CAPAM to CSC phenotype, we silenced PCIF1 in two CTC lines (Fig. S4b) and tested both SFP and resistance to chemotherapy. As expected, PCIF1/CAPAM knockdown reduced sphere number by 30% (Fig. 4e and Fig. S4c) as well as $m^6A_m$/A ratio while $m^6A$ level remained unaffected (Fig. S4d, e). Along the same vein, PCIF1 depletion in sh-FTO cells rescued the phenotype: sphere-forming ability (Fig. 4f and Fig. S4h) as well as chemosensitivity

(Fig. 4g). PCIF1 is also expressed at lower level in metastatic cell lines (Fig. S4g, h), concomitantly with reduced FTO expression (Fig. 3a). Balanced decrease of $m^6A_m$ writer and eraser (Fig. S4i) would explain the lack of change of $m^6A_m$ level with respect to cell lines originating from primary tumor (Fig. 4b).

Nevertheless, several lines of evidence suggest that FTO-mediated and PCIF1-mediated $m^6A_m$ dynamics cannot be merely

**Fig. 2 FTO silencing confers resistance to chemotherapy. a** Transient FTO silencing increases chemoresistance to FIRI. Graphs illustration of FIRI toxicity on either si-FTO or si-CTL-transfected cells (CRC1 (left) and SW620 (right) cell lines). Toxicity was measured using Sulforhodamine B assay. Bar plot represents quantification of IC50 of at least five biological replicates. FIRI 1 $X = 50\,\mu M$ 5-FU, $0.5\,\mu M$ SN38. Mean ± S.E.M, **$p$-value < 0.01, *$p$-value < 0.05, Two-sided unpaired $T$-test. **b** Stable FTO silencing increases chemoresistance to FIRI. Same as **a** with stable cell lines (sh-FTO or sh-CTL) from two distinct backgrounds (CRC1 (left) and SW620 (right)). Mean ± S.E.M, *$p$-value < 0.05, ns not significant, Two-sided unpaired $T$-test. **c** FTO silencing increases in vivo chemoresistance. Fifty-thousand of either sh-FTO or sh-CTL cells were subcutaneously injected into the flank of nude mice. After 21 days (tumor size about 100mm3), mice were treated with FIRI (50 mg FIRI per kg, 30 mg irinotecan per kg) and tumor growth was measured twice a week for 5 weeks. ***$p$-value < 0.001, two-way ANOVA test. Bar plot represent mean ± S.E.M of tumor volume measured at the last time point. **$p$-value < 0.01, Two-sided unpaired $T$-test. **d** FIRI-resistant cells display decreased FTO expression. Immunoblot analysis of METTL3–METTL14, ALKBH5 and FTO levels after 72 h of 0.2× FIRI treatment. Pictures are representative of three experiments in CRC1 cell line. Protein level quantification is mean ± S.E.M of FTO normalized to ACTIN of three biological replicates. *$p$-value < 0.05, ns not significant, Two-sided unpaired $T$-test.

viewed as two opposite poles of a two-sided enzymatic equation. First, while PCIF1 depletion negates sh-FTO phenotype, it does not affect basal CSC properties in CRC1 cells (Fig. S4f and S4g) that display lower basal stem-like abilities than CTC-derived cell lines. Second, PCIF1/CAPAM level decreases sphere-forming abilities but not resistance to FIRI treatment in CTC (Fig. 4e and Fig. S6a). Likewise, sublethal doses of FIRI which promote CSC phenotype (Fig. 2d) did not modify PCIF1 expression (Fig. S6b). Nevertheless, these cells displayed reduced FTO expression (Fig. S6b) as well as increased $m^6A_m$ level (Fig. S6c). This suggests that FTO and PCIF1 do not exhibit mere antagonistic activities but rather share partially overlapping enzyme substrate specificity. Distinct cellular distribution of these two effectors may explain such functional difference.

**FTO mediated $m^6A_m$ demethylation takes place in the cytoplasm.** While FTO sequence carries a nuclear localization signal (NLS), its cellular localization varies among several mammalian cell lines[42,43]. This spatial regulation may result in distinct substrate preference: a recent report shows that FTO catalyzes both $m^6A$ and $m^6A_m$ demethylation of mRNA in cytoplasm but targets preferentially $m^6A$ in cell nucleus[39]. In our cell lines (CRC1 and SW620), FTO is present in nuclear speckles[17] as well as in cell cytoplasm (Fig. 5a). We employed detergent-based cell fractionation procedure to separate nuclei from cytoplasm from sh-CTL and sh-FTO cells. The efficacy of this protocol was evaluated by immunoblot using cytoplasm-specific and nucleus-specific markers, respectively GAPDH and Histone H1 (Fig. 5b). Then, we extracted mRNA from both compartment and quantified $m^6A_m$ and $m^6A$ by LC-MS/MS (Fig. 5c). While this batch method introduces some degree of variability from one experiment to another in terms of the absolute difference in the mass-spectrometry measurements, the result was clearly consistent across biological replicates. We observed a significant increase (almost 3-fold) of $m^6A_m/A$ ratio in sh-FTO cytoplasm (in comparison with sh-CTL), while this ratio remained steady in the nucleus. By opposition, neither of these two cell compartments displays alteration of $m^6A/A$ ratio (Fig. 5c). This suggests that FTO-mediated $m^6A_m$ demethylation takes place in the cytoplasm in colorectal cancer cell lines.

Next, we evaluated FTO expression and localization in tumor microarrays (TMA) from different colorectal stages: adenoma, 1, 2, 3, 4, and metastases ($n = 52$). Global FTO expression did not show any significant change over the course of tumor evolution (Fig. 5d). Yet, subcellular distribution analysis unveiled interesting features. FTO expression is strictly nuclear in healthy adjacent tissue as well as in precursor lesions of CRC (adenoma, Fig. 5d). Then, following submucosal invasion (stage 1), FTO was also found in the cytoplasm (Fig. 5d). These observations suggest that the tumorigenic process alters subcellular FTO distribution and may impact its activity.

## Discussion

Our study addresses the specific function of FTO in recently established colorectal cancer cell lines with a focus on CSC phenotype. We show that decreased FTO activity plays a critical role in colorectal cancer by enhancing CSC properties including sphere forming, in vivo tumorigenesis, and chemoresistance. The underlying mechanism takes place in the cytoplasm and appears to involve demethylating $m^6A_m$ residues adjacent to the $m^7G$-cap in highly selected transcripts, most likely to adjust their translation efficiency.

As the initially characterized $m^6A$ demethylase, FTO has been studied in various types of cancers, often reported as a pro-oncogenic factor[44]. Inhibiting FTO in glioblastoma impairs self-renewal and cancer progression[45]. Targeting FTO decreases leukemic CSC self-renewal and sensitizes leukemia cells to T cell cytotoxicity[46]. FTO is highly expressed in some AML types, where it suppresses all-trans retinoic acid-induced cell differentiation and promotes oncogene-mediated cell transformation and leukemogenesis[44]. High levels of FTO are characteristic in cervical squamous cell carcinomas, where they promote resistance to chemo/radiotherapy and increases DNA damage responses[47]. More recently, high FTO expression was associated with lower survival rates in patients with breast cancer[48]. Finally, FTO promotes breast cancer cell proliferation, colony formation, and metastasis in vitro and in vivo[48]. By contrast, a recent study uncovered a tumor suppressor function of FTO in ovarian CSC[49]. On the same trend, our data support an anti-oncogenic role of FTO in colorectal cancer, emphasizing the importance of tissular context. This observation is also consistent with previous studies showing cancer-specific effects of $m^6A$ regulators. METLL3 activity promotes tumorigenesis in AML by enhancing BCL2 and PTEN translation[50], whereas in glioblastoma stem cells, it suppresses growth and self-renewal by reducing expression of ADAM19[45].

Interestingly, Kaplan–Meier survival analysis from cancer database shows that overall survival of colorectal cancer is higher with lower FTO. This is contradictory with our data that connects reduced FTO expression with enhanced chemoresistance and tumor initiation, as emphasized by our results from cell lines derived from CTCs, the source of lethal metastases. At least two parameters can account this inconsistency. First, we show that FTO expression in colon cancer cells is finely regulated at the post-transcriptional level, though the precise underlying mechanism remains to be determined. This observation is in agreement with a recent report in gastric tissue[38] and stresses the importance of quantifying gene expression at the protein level for diagnostic purpose. Second, as will be discussed below, FTO activity and substrate specificity may vary with its subcellular distribution. This parameter is even more critical in the context of colorectal cancer, where a fraction of FTO relocates from the nucleus to the cytoplasm at an early stage of the disease. Whether the cause

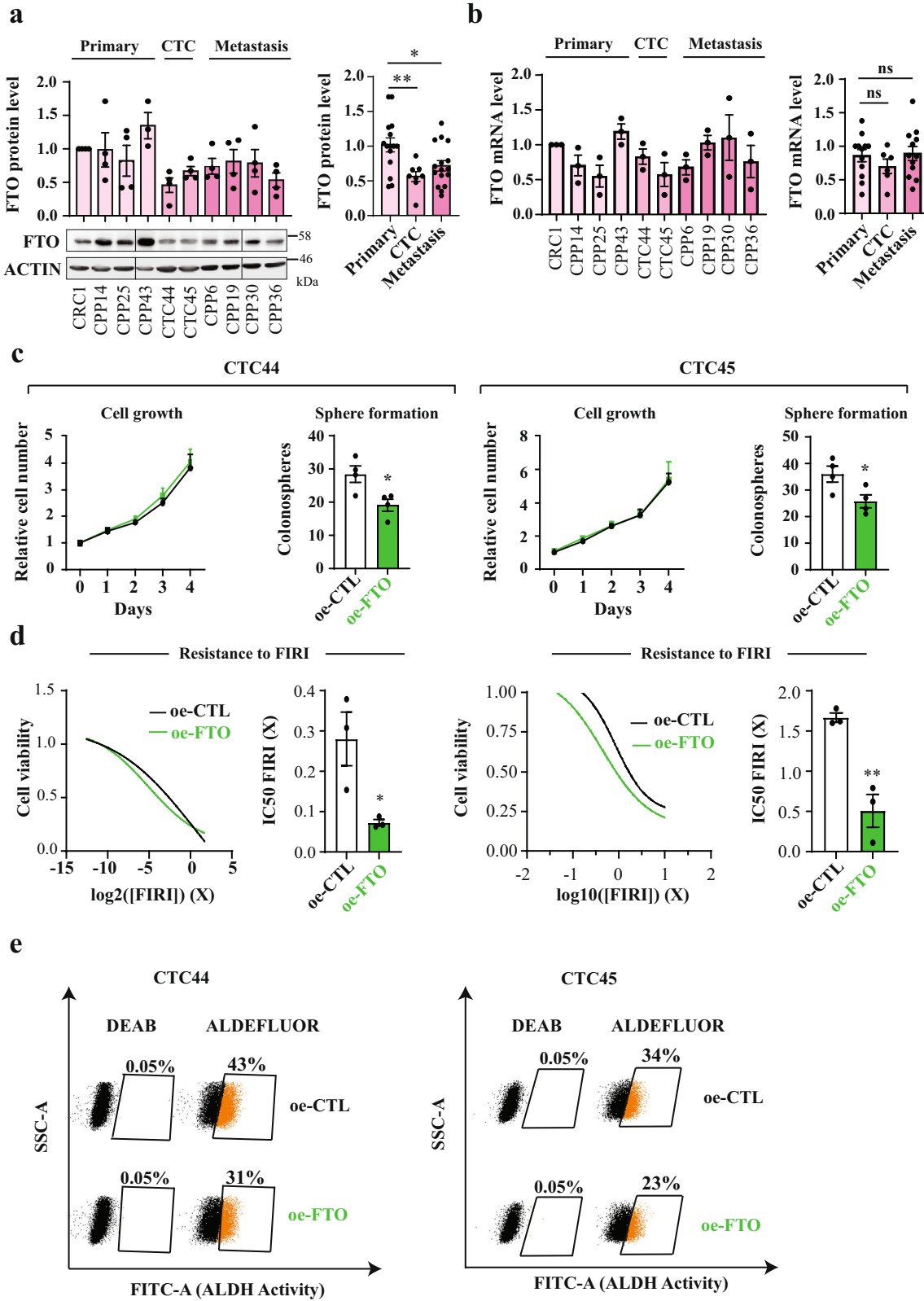

may be due to neoplastic transformation process remains to be determined.

Despite being ubiquitously expressed, FTO activity and function vary widely among tissues, as illustrated by transcriptome-wide mapping of m6A modifications[25,26]. To better comprehend the biological function of FTO, extensive efforts have been made to identify relevant RNA substrate(s). No clear consensus emerges

from recently published studies[29,39,51,52]. Following the identification of FTO as the first m6A mRNA demethylase[17], several reports connected demethylation of internal m6A nucleotides with a wide range of biological processes such as viral infection, stress-responses and DNA UV damage-responses[53–55]. More recently, Mauer et al. threw a stone into the pond by suggesting that FTO preferentially demethylates m6Am residues closely

**Fig. 3 FTO expression is lower in "circulating tumor cells" (CTC) than in "solid" tumor cells. a** FTO protein level is the lowest in Circulating Tumor Cell lines. FTO protein level quantification by western blot in several patient derived cell lines from distinct origin: primary tumor, blood, and metastasis. The first graph represents the level of FTO protein in each individual cell lines. The following bar plot represents the quantification by groups. Mean ± S.E.M of four biological replicates, **p-value < 0.01, *p-value < 0.05, one-way anova followed by multiple comparison. **b** FTO mRNA level barely fluctuates throughout colon cancer progression. Level of FTO mRNA was evaluated by RT-PCR in the same cell lines as in **a**. Left graph represents of FTO level in each individual cell line. Right bar plot represents the quantification by groups. Mean ± S.E.M of three biological replicates, ns not significant, One way anova. **c** Increase of FTO level decreases the sphere forming potential of CTC cell lines. Cell growth and sphere forming potential evaluation after overexpression of FTO in CTC44 and CTC45 lines. **d** Increase of FTO level sensitizes CTC cell lines to chemotherapy. FIRI toxicity assay on CTC44 or CTC45 after overexpression of FTO. Mean ± S.E.M, **p-value < 0.01, *p-value < 0.05, Two-sided unpaired T-test. **e** FTO overexpression decreases ALDH activity in CTC cell lines. Flow cytometry quantification of ALDH positive cells after FTO overexpression. The number of ALDH positive cells in one representative biological replicate out of three for CTC44 and CTC45.

adjacent to the cap[29]. The higher in vitro demethylation efficiency reported was confirmed by Wei et al., although the conclusion was moderated in the cellular context, since the level of internal m6A marking is ten-fold higher than m6Am marking[39]. Discrepancies between studies on FTO may arise from several known biological parameters. A recent review addresses this ambiguity and emphasizes the context-dependent functions of RNA methylation effectors[56]. First, FTO displays a wide substrate spectrum: besides mRNA (and tRNA for m1A[39]), FTO can demethylate m6A and m6Am in snRNAs[51]. Second, its spatial distribution dictates substrate preferences: cytoplasmic FTO catalyzes demethylation of both m6A and cap-m6Am whereas nuclear FTO demethylates preferentially m6A[39], most likely owing to accessibility constraints to the cap-moiety. In colorectal cancer cell lines, FTO activity displays a remarkable selectivity for cap-m6Am, which concurs with its presence in the cytoplasm. LC-MS/MS analysis of small RNA species did not reveal any change of m6A and m6Am levels following FTO silencing. Also, our bioinformatics analysis of sh-FTO vs. sh-CTL RNA-seq data failed to identify the significant alterations of mRNA splicing that might result from altered snRNA methylation. While we cannot exclude the subtle alteration of internal m6A marks that would have been missed out due to the limit of detection of m6A changes by MeRIP-seq[57], our data clearly establishes cap-m6Am residues of mRNA as the main substrate of cytoplasmic FTO in colorectal cancer cells. Extending this observation, targeting either the m6A writer complex or ALKBH5 does not significantly affect CSC phenotype. By contrast, inhibiting PCIF1/CAPAM offsets the inhibiting effect of FTO overexpression on the CSC phenotypes. PCIF1/CAPAM catalyzes cap-Am methylation in the nucleus while FTO demethylates m6Am in the cytoplasm. Compartment-specific enzymatic activity implies partially overlapping mRNA substrates and explains why m6Am writer and eraser do not display mere antagonistic activities.

In this study, we clearly identify m6Am as a critical epitranscriptomic mark for controlling the stem-like cell phenotype of human colon cancer cells. While m6Am was first identified in mRNA from mammalian cells and viruses in the 70s[27], the enzymes catalyzing m6Am modifications, PCIF1/CAPAM and FTO, were only recently identified, and the study of m6Am role in mRNA metabolism and cellular function is still in its infancy. Due to its abundance and its strategical location in the cap structure[58], this chemical modification holds an inherent potential in gene expression control. Mauer et al. reported that FTO-catalyzed m6Am demethylation reduced mRNA stability by rendering it more vulnerable to DCP2-mediated decapping process[29]. To investigate this possibility in CRC, we performed whole transcriptome sequencing (RNA-seq, 125 bp, paired-end, and $n = 3$) of sh-FTO and sh-CTL cells. The volcano plot of the transcriptomic variation between sh-CTL and sh-FTO cells revealed minor changes at the transcriptome level (Fig. S7a, Table S4, and Supplementary data 1). Such minor differences in mRNA levels suggested that FTO may act in CSCs by modulating translation efficiency of individual mRNAs. This scenario is consistent with the proximity of m6Am adjacent to the m7G-Cap, as well as a recent study on PCIF1 suggesting a role of m6Am modification in translational control of certain mRNAs[28]. We therefore performed RNA-seq on mRNA co-sedimenting with heavy polysomes (4+ ribosomes/mRNA) or light polysomes/monosomes (1–3 ribosomes per mRNA). We mapped reads to the human reference transcriptome to determine mRNAs that were differentially translated between sh-CTL and sh-FTO cell lines. The volcano plots of the light and heavy analyses (Fig. S7b, c, Tables S5 and S6, and Supplementary data 1) did not show profound changes at the translational level. This suggests two possibilities: either m6Am-mediated alteration of gene expression necessitates a stress (e.g., suspension culture or chemo-treatment) or another molecular mechanism, disconnected from gene expression control, is involved.

In summary, we have shown that cytoplasmic FTO activity regulates m6Am modification of selected mRNAs and, subtly but surely, is necessary for maintaining the CSC phenotype in vitro and in vivo for human colon cancers. The extent to which this applies to other cancers remains to be determined: given the complexity of cancer, there will likely be a great variability between cancer types. Even if limited to colon cancer, however, our findings point the way to deeper understanding of the CSC phenotype and have exciting implications for developing therapies that decrease m6Am modification to cripple CSC-based metastases and resistance.

## Methods

**Cell lines**. Patient-derived colon cancer cell lines (CRC1, CPP-14/25/43/6/19/30/36) were derived from CRC surgeries provided by CHU-Carémeau (Nîmes, France, ClinicalTrial.gov Identifier#NCT01577511) within an approved protocol by the french Ethics Committee: CPP (Comité de Protection des Personnes) Sud Méditerrannée III. We have complied with all relevant ethical regulations for work with human participants, and informed written consent was obtained for all the patients. CRC1, CPP-14/25/43 cell lines were derived from primary tumors and CPP-6/19/30/36 from metastatic tumors. CTC44 and CTC45 are circulating tumor cell lines derived from blood of metastatic chemotherapy-naïve stage IV CRC patients[31]. HCT-116 (ATCC® CCL-247™) and SW620 (ATCC® CCL-227™) are commercially available colon cancer cell lines derived, respectively from primary and metastatic human tumors.

**Antibodies**. A list of antibodies used in this study is provided in supplementary section (Table S2) with dilutions and references for commercial antibodies.

**Cell culture and generation of stable cell lines**. Cells were maintained at 37 °C under humidified 5% $CO_2$ in DMEM medium (Gibco) supplemented with 10% FCS (Eurobio) and 2 mM glutamine. Stable knockdown of FTO was achieved by lentiviral delivery (5 D.O.I) of anti-FTO sh-RNA (Origene, #TL308064, sh-FTO#B). Isolation of infected cells was performed by GFP positive cells sorting on FACSAria.

**Plasmid constructions**. FTO coding sequence was amplified from pDONOR plasmid (Montpellier Genomic Collection MGC Facility) and inserted into pCDNA3-Flag Cter plasmid into HindIII and XbaI restriction sites.

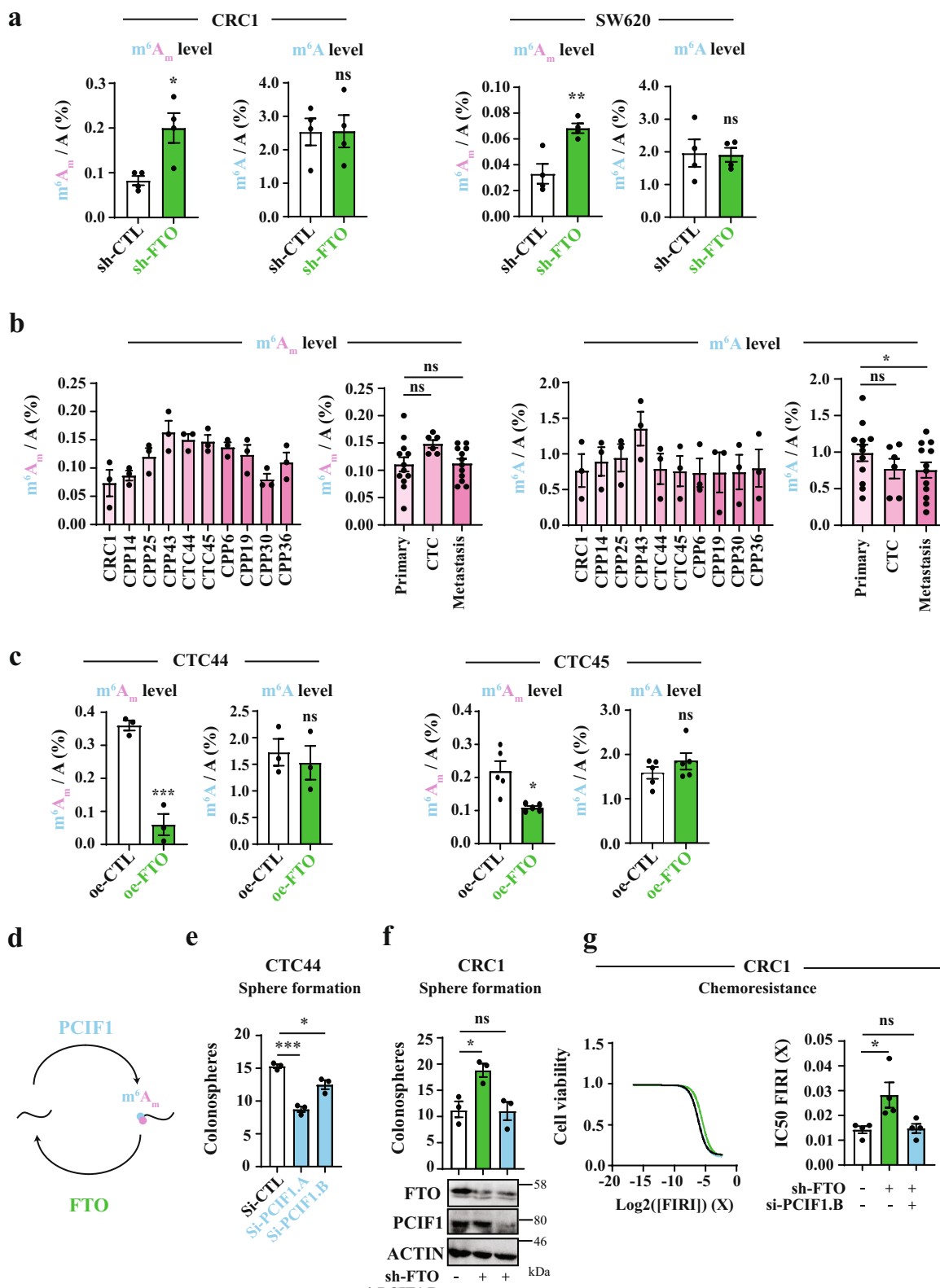

**Transfections**. Transfection of 100 nM of si-RNA duplex was performed using Lipofectamine RNAimax (Invitrogen) according to the manufacturer's instructions. The sequences of siRNA are provided in Table S3. 300,000 cells in 6-well plate were transfected for 48 h with 2 µg of plasmid DNA using Lipofectamine 2000 (Invitrogen) according to the manufacturer's instructions.

**FIRI treatment**. Cells were treated for 72 h at 37 °C under humidified 5% CO2 with 10 µM of 5-fluorouracile coupled with 0.1 µM of SN38.

**RNA extraction and RT-qPCR**. Total RNA was extracted using TRIzol reagent (Invitrogen) according to the manufacturer instructions. For RT-qPCR analyses, 1 µg of RNA was reverse-transcribed into cDNA using random hexamer (Invitrogen) and 1 U of MML-V reverse transcriptase (Invitrogen). Quantitative gene expression was performed using SYBR Green master mix (Roche) on LightCycler 480 Instrument (Roche). Results were normalized to actin expression and analyzed using the ΔΔCt method. Primer sequences are provided in Table S1.

**Fig. 4 FTO affects stem-like properties through m$^6$A$_m$ demethylase activity. a** FTO silencing increases m$^6$A$_m$ level rather than m$^6$A. LC-MS/MS mRNA quantification of m$^6$A$_m$/A and m$^6$A/A ratios in either CRC1 sh-FTO or sh-CTL and SW620 sh-FTO or sh-CTL cell line. Bar plot represents mean ± S.E.M of at least three biological replicates. **p-value < 0.01, *p-value < 0.05, ns not significant. Two-sided unpaired T-test. **b** m$^6$A$_m$ tends to be increased in CTC lines but not m$^6$A. LC-MS/MS analysis of mRNA from patient derived cell lines (same as in Fig. 3). m$^6$A$_m$/A and m$^6$A/A were evaluated. Graphs represent mean ± S.E.M of m$^6$A$_m$/A or m$^6$A/A level for each cell line or group of cell lines. Three biological replicates. **c** FTO overexpression decreases m$^6$A$_m$ rather than m$^6$A in CTC lines. mRNA quantification of m$^6$A$_m$/A and m$^6$A/A after FTO overexpression in CTC44 line. Graphs represent mean± S.E.M of at least three biological replicates. ***p-value < 0.001, *p-value < 0.05, ns not significant. Two-sided unpaired T-test. **d** Effectors of m$^6$A$_m$ modification. **e** PCIF1 silencing decreases colonosphere formation. Sphere forming ability of CTC44 line after silencing of the m$^6$A$_m$ writer PCIF1. Bar plot represents mean ± S.E.M of three experiments. ***p-value < 0.001, *p-value < 0.05. Two-sided unpaired T-test. **f** PCIF1 silencing rescues sphere forming ability in CRC1 sh-FTO cell line. PCIF1 was depleted by siRNA treatment in sh-FTO cell line and sphere formation assay was performed. Results are mean ± S.E.M of three independent experiments. *p-value < 0.05, ns not significant. One-way Anova followed by multiple comparisons. **g** PCIF1 silencing rescues chemosensitivity in CRC1 sh-FTO cell line. FIRI toxicity was measured with or without siRNA-mediated PCIF1 silencing in sh-FTO cell line. Cell viability curve of one representative experiment is shown (left). Results are expresses as mean ± S.E.M (barplots) of four independent experiments (right). *p-value < 0.05, ns not significant. One-way anova followed by multiple comparisons.

**Protein extraction and western blot**. Cells were washed twice with ice cold phosphate-buffered saline (PBS) and lysed in RIPA buffer (50 mM Tris, 150 mM NaCl, 1% NP-40, 0.25% sodium deoxycholate, 2 mM sodium orthovanadate, 50 mM Sodium fluoride, 50 mM β-glycerophosphate, 2 mM EGTA; pH 7.5). Samples were separated on a 12% SDS-polyacrylamide gel electrophoresis, transferred to nitrocellulose membrane, blocked for 1 h in 5% (w/v) non-fat dry milk in PBS and probed overnight at 4 °C with primary antibodies (Table S2). Membranes were incubated with secondary antibody for 1 h and proteins were revealed by ECL Prime (Amersham) using ChemiDoc Touch imager (Biorad).

**Proliferation assay**. A thousand cells were seeded in triplicate in 96-well plate for 24, 48, 72, and 96 h. Then, cells were fixed for at least 2 h at 4 °C in 10% TCA. After three washes with Milli-Q water, cells were incubated in 0.4% Sulforhodamine B solution for 30 min at room temperature followed by three washes with MilliQ water. Absorbance was measured at 562 nm after resuspension in 10 mM Tris-Base.

**Sphere formation assay**. This test was performed as previously described[59]. Number of sphere forming cells were determined after plating of 100 cells/100 μl of M11 medium (DMEM/F12 (1:1) Glutamax medium, N2 Supplement, Glucose 0.3%, insulin 20 μg/ml, hBasic-FGF 10 ng/ml, and hEGF 20 ng/ml) in ultra-low attachment 96 well-plates. Spheres > 50 μm were counted after 5–7 days.

**Cytotoxicity assay**. Two thousand cells were seeded in a 96-well plate. After 24 h, cells were treated with decreasing doses of 5-fluorouracile (5-Fu) coupled with SN38 (FIRI) for 72 h (1/3 dilution from 3.3 X to 0 X; 1 X = 50 μM 5-FU + 0.5 μM SN38). FOX toxicity was assessed with decreasing doses of 5-fluorouracile coupled with oxaliplatin for 72 h (1/3 dilution from 3.3 X to 0 X; 1 X = 50 μM 5-FU + 1 μM oxaliplatin). Cell viability was measured using Sulforhodamine B assay as previously described and IC$_{50}$ was determined graphically.

**Flow cytometry**. The ALDH activity of adherent cells was measured using the ALDEFLUOR kit (Stem Cell Technologies), according to the manufacturer's instructions. CD44 and CD44v6 were stained using anti-CD44 antibody and anti-CD44v6 antibody for 15 min at 4 °C (Table S2). As a reference control, anti-IgG2a and REA-S control isotype was used under identical conditions. The brightly fluorescent ALDH, CD44, or CD44v6 positive cells were detected using a MACSQuant Analyzer (Miltenyi Biotec). To exclude nonviable cells, Sytox blue was added at a concentration of 0.1 μg/ml.

**Immunofluorescence**. Cells were fixed in PBS containing 4% paraformaldehyde at room temperature for 15 min, wash twice in PBS, permeablized with 0.1% NP-40 in PBS for 10 min, wash twice in PBS and blocked with 5% FCS for 30 min. Coverslips were incubated 1 h with primary antibody (Phosphosolution, 597-FTO) at RT. After washing three times with PBS, coverslips were incubated for 1 h with Alexa Fluor®-conjugated secondary antibody (Alexa Fluor® 488 Goat Anti-Mouse (IgG), Invitrogen) at RT. For nuclei staining, coverslips were washed twice and incubated with 1 μg/ml Hoechst 3358 for 5 min at RT. After two washes with distilled water, coverslips were mounted on slides with Fluoromount-G (Invitrogen). Fluorescent pictures were acquired at room temperature on an AxioImager Z1 microscope (Carl Zeiss, Inc.) equipped with a camera (AxioCam MRm; Carl Zeiss, Inc.) and Plan Apochromat (63×, NA 1.4) objective, the Apotome Slider system equipped with an H1 transmission grid (Carl Zeiss, Inc.), and Zen 2 imaging software (Carl Zeiss, Inc.).

**mRNA purification**. mRNA was purified from total RNA with two rounds of GeneElute mRNA purification kit (Sigma). rRNA was removed using Ribominus kit (Invitrogen) according to the manufacturer's instructions.

**Nucleoside mass-spectrometry analysis**. This part was performed as previously described[39]. Briefly, 400 ng of RNA was digested by 5 U of RppH (New England Biolabs) for 2 h at 37 °C. Decapped mRNA were then digested by 1 U of Nuclease P1 (Sigma) for 2 h at 42 °C in NH$_4$OAc buffer (10 mM, pH 5.3). Nucleotides were dephosphorylated for 2 h at 37 °C by 1 U of Alkaline phosphatase in 100 mM of NH$_4$OAc. The sample was then filtered (0.22 μm pore size, 4 mm diameter, Millipore), and 10 μl of the solution was injected into LC-MS/MS. The nucleosides were separated by reverse phase ultra-performance liquid chromatography on a C18 column with online mass spectrometry detection using Agilent 6490 triple-quadrupole LC mass spectrometer in multiple reactions monitoring (MRM) positive electrospray ionization (ESI) mode. The nucleosides were quantified by using the nucleoside-to-base ion mass transitions of 282.1 to 150.1 (m$^6$A), 268 to 136 (A), 296 to 150 (m$^6$A$_m$), and 282 to 136 (A$_m$).

**Tumor initiation assay**. Decreasing amount of cells (1000, 500, and 100) were subcutaneously injected into nude mice (Hsd:Athymic Nude-Foxn1nu nu/nu, 6 weeks, females, five mice per group) in Matrigel-DMEM (v: v). Tumor sizes were measured twice a week for 50 days. After 50 days, the mice were sacrificed and tumors were taken out. The number of mice bearing growing tumor (size > 100 mm$^3$) was counted. Tumor apparition frequency was determined using online ELDA (extreme limiting dilution analysis) software (https://bioinf.wehi.edu.au/elda/software).

**Chemoresistance in vivo**. Fifty thousand cells were subcutaneously injected into nude mice in Matrigel-DMEM (v: v). 50 mg/kg 5-FU + 30 mg/kg Irinotecan treatment (two i.p injection a week), was initiated once tumor reached 100 mm$^3$ [60]. These studies complied with all relevant ethical regulations for animal testing and research. They were approved by the ethics committee of the Languedoc Roussillon Region and carried out in compliance with the CNRS and INSERM ethical guidelines of animal experimentation (CEEA-LR-12051).

**Purification of nucleus and cytoplasm**. Cells were harvested and washed twice in ice-cold PBS. Then, cell pellet was resuspended in fractionation buffer (0.5% NP-40, 150 mM NaCl, 50 mM Tris-HCl pH 7.4) and left on ice for 3 min. Cytoplasmic fraction was separated from nucleoplasmic fraction by centrifugation at 16,000 × g for 10 min at 4 °C. The supernatant containing the cytoplasmic fraction was collected and cell debris were removed by an additional centrifugation at 16,000 × g for 10 min at 4 °C. Nuclear pellet was washed twice in wash buffer (150 mM NaCl, 50 mM Tris-HCl pH 7.4) and resuspended in fractionation buffer.

**Tissue microarray (TMA)**. TMA was constructed with FFPE tumor samples collected in the frame of the Clinical and Biological Database BCBCOLON (Institut du Cancer de Montpellier—Val d'Aurelle, France, Clinical trial Identifier #NCT03976960). Adenomas, primary adenocarcinomas, and metastatic lesions were sampled as two cores of 1 mm diameter. All samples were chemonaive. Tumor samples were collected following French laws under the supervision of an investigator and declared to the French Ministry of Higher Education and Research (declaration number DC-2008–695). Study protocol has been approved by the french Ethics Committee: CPP (Comité de Protection des Personnes) Sud Méditerrannée III (Ref#2014.02.04) and by the local translational research committee (ICM-CORT-2018-28). We have complied with all relevant ethical regulations for work with human participants, and informed written consent was obtained for all patients.

**FTO detection by immunohistochemistry**. Three-micrometer thin sections of formalin-fixed paraffin-embedded tissues were mounted on Flex microscope slides (Dako) and allowed to dry overnight at room temperature before immunohistochemistry processing, as previously described[61]. Briefly, PT-Link® system (Dako)

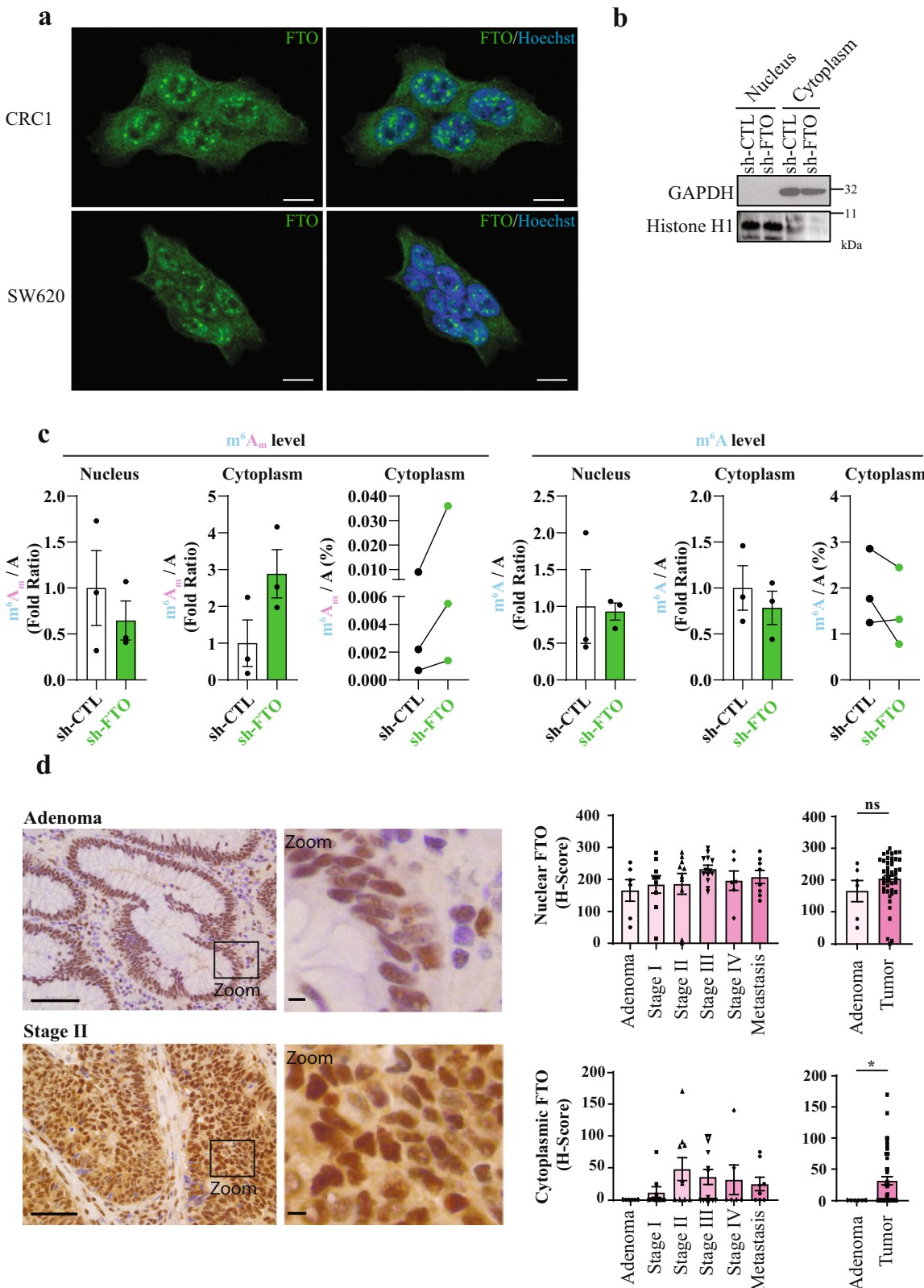

was used for pre-treatment. Then, heat-induced antigen retrieval was executed for 15 min in High pH Buffer (Dako) at 95 °C. Immunohistochemistry procedure was performed using the Dako Autostainer Link48 platform. Endogenous peroxidase was quenched using Flex Peroxidase Block (Dako) for 5 min at room temperature. Slides were then incubated with the anti-FTO rabbit monoclonal antibody (Abcam, Clone EPR6895; 1/1000) for 20 min at room temperature. After two rinses in buffer, the slides were incubated with a horseradish peroxidase-labeled polymer coupled to secondary anti-mouse and anti-rabbit antibodies (Agilent, EnVision

FLEX #K8000) for 20 min, followed by appliance of 3,3′-Diaminobenzidine for 10 min as substrate. Counterstaining was performed using Flex Hematoxylin (Dako) followed by washing the slides under tap water for 5 min. Finally, slides were mounted with a coverslip after dehydratation.

Two independent observers analyzed the TMA slide in a blinded manner. The semiquantitative *H*-score method[62] was used to convert the expression of FTO to continuous values, based on both the staining intensity and the percentage of cells at that intensity. Nuclear and cytoplasmic signals were taken

**Fig. 5 FTO mediated m6Am demethylation takes place in the cytoplasm. a** FTO localizes both in nucleus and cytoplasm. Immunofluorescence staining of FTO (green) in CRC1 and SW620 cell lines show presence of FTO in cytoplasm and in nuclear speckles. Nucleus of cells were stained with Hoechst (blue). Pictures are representative of several fields and two biological experiments. Scale bar 10 μm. **b** Verification of the effective cell fractionation procedure by immunoblot. Effective separation of cytoplasmic and nuclear fractions was evaluated by immunoblot using cytoplasmic marker (GAPDH) and nuclear marker (Histone H1). **c** FTO silencing increases cytoplasmic m6Am level. mRNA quantification of m6Am/A and m6A/A level for nuclear fraction and cytoplasmic fraction. Bar plots represents mean ± S.E.M of three biological replicates. Before–After plots represents the same data as bar plot with raw values. **d** FTO relocalizes to the cytoplasm during tumorigenesis. FTO expression and localization was evaluated by IHC on TMA from CRC patient. Then, nuclear and cytoplasmic level of FTO were quantified. Pictures are representative of stage 0 and stage 2. Scale bars 100 μm and 10 μm (zoom). Bar plots represents mean ± S.E.M of $H$-score based on nuclear and cytoplasmic intensity of FTO staining. Each dot corresponds to individual value. *$p$-value < 0.05, ns not significant, two-tailed Mann–Whitney test.

into account separately. Staining intensity was scored as no staining (0), weak staining (1), moderate staining (2), or intense staining (3). The percentage of cells stained at certain intensity was determined and multiplied by the intensity score to generate an intensity percentage score. The final staining score of each tissue sample was the sum of the four intensity percentage scores, and these scores ranged from 0 (no staining) to 300 (100% of cells with intense staining). In all cases with discrepant results, a consensus was reached between both investigators. Averaged FTO $H$-score was given when both cores from a single sample were assessable.

**Reporting summary**. Further information on research design is available in the Nature Research Reporting Summary linked to this article.

## Data availability
The data discussed in this publication have been deposited in NCBI's Gene Expression Omnibus[63] and are accessible through GEO Series accession number GSE165115. Source data are also provided with this paper. All other relevant data are available from the corresponding author on reasonable request. Source data are provided with this paper.

## Code availability
Data analysis was performed using free software detailed in the material and methods (i.e. FastQC v0.11.5, SortMeRNA v2.1b, Kallisto v0.45.0 and R v3.5.1). Statistics and graphics were performed using R packages (DESeq2, ggplot2, biomaRt) and an online program gProfileR. All the scripts used are hosted on a private gitlab repository and could be available from the corresponding author on reasonable request (contact: rivals@lirmm.fr).

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

## Acknowledgements
This work was generously supported by Occitanie Region/FEDER (PPRi, SMART project), Ligue contre le Cancer, SIRIC Montpellier Cancer (INCa-DGOS-Inserm 6045), Cancéropole GSO and Labex NumeV (GEM flagship project, ANR 2011-LABX-076). We thank Plateforme de Proteomique Clinique (PPC, https://ppc-montpellier.com), Montpellier Genomix (http://www.mgx.cnrs.fr) sequencing facility as well as iExplore animal facility, in particular Denis Greuet and Steeve Thirard. We thank the ATGC bioinformatic platform, whose servers hosted our bioinformatic analyses; ATGC is a member of both the "France Génomique" network [ANR-10-INBS-0009] and the Institut Français de Bioinformatique [ANR-11-INBS-0013].

## Author contributions
A.D., A.B., E.R., and S.R. designed experiments and analysed the results. S.R., H.G., A. Am., C.A., F.B., A.At., J.V., V.M., F.D., A.C., and F.M. performed experiments. A.D., A.B., E.R., S.R., C.H., E.C., J.J.V., E.S., Y.M., F.A., and J.P. performed data analyses. J.R., S.R. and E.R. designed bioinformatics pipelines and performed bioinformatics analysis. A.D., A.B., E.R., and S.R. wrote the manuscript. All the authors reviewed the final version of the manuscript.

## Competing interests
The authors declare no competing interest.
