## [Peer Review File · Nature Communications]

Reviewers' comments:

Reviewer #1 (Remarks to the Author):

In this study, Relier et al. showed that the FTO impedes CSC abilities in colorectal cancer through its m6Am demethylase activity. Low FTO expression in patient-derived cell lines elevates m6Am level in mRNA which results in enhanced in vivo tumorigenicity and chemoresistance. Inhibition of the m6Am methyltransferase of PCIF1/CAPAM partially reverses this phenotype. The authors showed that FTO/m6Am axis constitutes a reversible pathway controlling CSC abilities that does not involve transcriptome remodeling, but rather modulates translation efficiency of selected m6Am marked transcripts. This is an interesting work. But the manuscript need be improved to clarify the following questions.

1. It is somehow strange that the knockdown of FTO will not lead to the change of m6A in mRNA. The authors need to carefully measure the level of m6A in control and FTO knockdown samples.
2. In Figure 1 and 2, the authors found that FTO silencing promotes stem-like traits in colorectal cancer cell lines and confers resistance to chemotherapy by siRNA or sh-FTO knockdown. I would suggest the authors to perform parallel experiments by overexpressing FTO and then examine the effect upon FTO overexpression.
3. The authors had better provide evidences to demonstrate the purity of isolated mRNA for detecting the modifications of m6Am, m6A, and Am since other RNA species also carry these modifications. Contamination of other RNA species in mRNA will cause the inaccurate quantification of tested modifications in mRNA, which may then lead to the wrong results.
4. As for the mass spectrometry measurement, the normal way to express the levels of these modifications is in the format of percentages of m6A/A, m6Am/A, Am/A. In this way, the authors can compare the measured levels of these modifications to previously reported levels of these modifications, which will also show the accuracy of the measurement. Right now the authors used the fold changes and the normalized data. With the current information, people don't know the real levels of these modifications. Moreover, I would suggest the authors to use m6Am/A, instead of m6Am/Am, to demonstrate the connect between m6Am and FTO.
5. The author showed that low FTO expression in patient-derived cell lines elevates m6Am level in mRNA which results in enhanced in vivo tumorigenicity and chemoresistance. However, in the summary, the authors mentioned that "Even if limited to colon cancer, however, our findings point the way to deeper understanding of the CSC phenotype and have exciting implications for developing therapies that increase m6Am modification to cripple CSC-based metastases and resistance". These two are contradictory.

Reviewer #2 (Remarks to the Author):

I enjoyed reading this manuscript. The authors have performed quite comprehensive investigations of roles of RNA m6A regulators on colorectal cancer stem cells. They discovered that FTO is the only one that seems to affect CSC stem-like properties and resistance to chemo treatment. This part of the work is vigorous and well done. However, the main conclusion and the title are all incorrect or misleading. After reading this manuscript my conclusion was the role of FTO has less to do with mRNA modifications. The authors showed that the m6A level did not change much but cap m6Am level showed substantial change. These results are also misleading as they did not label correctly the exact percentage of m6Am/A and the change. Please do not show fold change in y-axis. A modification could change from 0.0001% to 0.0003% showing 3-fold change but has no functional relevance. If they label the exact level of m6Am/A the ratio is not going to be impressive.

My biggest problem is that in the title and discussion the authors conclude that m6Am axis controls the cancer stem-like property of colorectal cancer. To justify they need to perform all functional

characterization of PCIF1/CAPAM which installs the cap m6Am. This part is very thin and data not consistent at all. In fact PCIF1/CAPAM should have much stronger phenotype if the authors conclusion is correct. This is clearly not the case.

The correct way to present the data is to analyze cellular localization of FTO first, measure nuclear versus cytoplasm m6A/A and m6Am/A levels (show exact percentage), suggesting that these changes may have minor effects on CSC as i) METTL3/14 KD did not affect CSC stem like properties; ii) PCIF1 KD could not recapitulate most CSC property changes observed with FTO KD. In this particular system FTO is likely working on a particular group of RNA m6A. This should be pursued in the future.

Note that even the translation changes observed by the authors attributing to m6Am seem to be minor. I doubt the authors can use these translation differences to recapitulate phenotypes observed with FTO KD. It is perfectly fine to discover nice phenotypes and that none of the current pathways could explain the phenotypes suggesting new knowledge to be learned.

Note in other systems KD FTO reduces cell growth (HeLa, HEK, AML etc. etc). The authors need to be careful here. It seems to be dependent on targets of FTO.

Note that the overall survival of colorectal cancer is higher with lower FTO (check cancer database)! This is not consistent with shFTO promoting tumor initiation. The authors need to discuss this part.

In addition, see drug resistance in other systems: <https://www.nature.com/articles/s41422-018-0097-4>

Other questions:

Fig. 2f. It looks to me the protein levels of METTL3 and METTL14 both increased and ALKBH5 showed decrease.

Fig. 4C is not consistent with Fig. 3C, note that m6Am did not increase while FTO level reduced by 50%! This essentially suggested that m6Am is not the target. And please show the absolute percentage. What is the level of PCIF1? This again indicates m6Am is not the source.

The authors need to show KD PCIF1 in shFTO cells rescue phenotypes. Not a single experiment has been shown.

I suggest the authors perform MeDIP on m6A and see if they identify any unique targets.

The authors should analyze translation efficiency (protein level/mRNA) and seek correlation with m6A or m6Am changes. To me it could be a specific group of RNA. Their Fig. 6a suggested m6A. The Wei ref is data from HEL cells. The authors should perform m6A MeDIP from their cells and preform analysis again.

Lastly, if PCIF1 is involved can authors search cancer database and show its level correlation with survival and compare with FTO? I bet PCIF1 has no role in colon cancer.

Reviewer #3 (Remarks to the Author):

In this study, Relier et al investigated the role of mRNA m6Am modification on colorectal cancer stem cell (CSC) function. They focused on FTO, which has RNA demethylase activity and show that knockdown or overexpression of FTO alters the ability of CRC to grow as spheres in 3D and response to chemotherapeutic drug combinations, in vitro and in mouse xenografts. Overall, although the phenotypic data looks good and convincing for the most part, the role of regulation of mRNA

translation by FTO is a major concern. In addition, the conclusions are overstated and not supported by the data. Specific comments/concerns are:

1. It is concerning that in the transcriptome analysis from shCTL vs shFTO cells, FTO expression does not change significantly (Table S4). This means that the knockdown of FTO was not sufficient and the conclusion on line 183 that FTO mainly acts at the level of translation may not be true. In fact, the authors may be missing out important targets of FTO that could help explain the observed phenotypes. This is a major concern. If the authors choose to address this, they should repeat the knockdown experiment and perform total RNA-seq. This should be done in two CRC lines and the genes that are commonly up- or down need to be selected for further analysis to try to see if they can explain the observed phenotype.

2. The polysome data presented in the current manuscript does not explain the observed phenotypes. It could be that key genes are missed out due to incomplete knockdown of FPO. If shRNAs don't work very efficiently, the authors can knockout FTO using CRISPR/Cas9. HCT116 has been known as an excellent cell line for genome editing. The polysome experiments will have to be repeated to first make sure that the knockdown of FTO was substantial, followed by analysis of mRNAs shoes translation is deregulated. One would expect that this approach may be useful in identifying the genes regulated by FTO that are important for its biological function.

3. In the sphere forming data in Figure 1C and 1D, for example, there error bars in the si-CTL are missing. Without error bars in the si-CTL, how did the authors calculate the p-value? This is a problem in majority of the Figures where the data for the control is set to 1. There are no error bars in the control.

4. Figure 2C: It will be good to show the tumor volume for the last time point for the sh-CTL and sh-FTO.

5. The authors show that chemoresistance cells have 50% less FTO protein. Is this due to less FTO mRNA levels? This can be determined by qRT-PCR.

6. Line 123: What was the sub-lethal dose used in this experiment?

7. Figure 1f: How could the authors calculate p-values without errors bars in the control sample (DMSO). Looks like METTL3 levels were substantially reduced upon FIRI treatment and ALKBH5 went up.

8. The authors show that FTO expression is linked to chemoresistance. FTO kd results in increased tumor growth in response to CRC chemotherapy in vivo (Figure 2). This suggests that FTO inhibits tumor growth in response to chemotherapy. Would you not expect FTO levels to increase in response to chemotherapy, in contrast to what the authors show where they found that when CRC1 cells are treated with sub-lethal dose of chemotherapy, FTO levels decrease. Also, what was the dose used for Figure 2f? Can the authors perform PI staining and FACS analysis at this dose so that the effect on cell cycle and sub-G1 population is clear?

9. Figure 3d: It is unclear how the FTO protein levels in the Primary samples is not 1? How was this data normalized?

10. The data in Figure 3 can be presented in a better way. In the text, the authors start with Figure 3c and 3d. Why not make these panels Figure 3a and 3b?

11. The conclusion in for the data in Figure 4 is overstated. The authors show that modulating FTO levels, alters m6Am/m6A ratio but these results are from a single cell line, CRC1. One would expect that this cell line has a high proportion of CSCs, but it still is a cell line, not a population of CRC1 cells

enriched for CSCs.

12. Line 105: change $p=0,00035$ to $p=0.00035$

13. Line 117: To evaluate "the effect" of FTO....

14. Line 115, change "inhibition" to "knockdown" or "silencing". Inhibition is used for inhibitors, not shRNAs.

15. Line 146: Please specify in the text which RNase was used.

We are grateful to the reviewers for their insightful and helpful comments to improve significantly our manuscript NCOMMS-19-379072 now entitled "**FTO-mediated cytoplasmic m⁶A_m demethylation adjusts stem-like properties in colorectal cancer cell**".

Please, find below in blue text our point-by-point response to the reviews.

Reviewers' comments:

Reviewer #1 (Remarks to the Author):

In this study, Relier et al. showed that the FTO impedes CSC abilities in colorectal cancer through its m⁶A_m demethylase activity. Low FTO expression in patient-derived cell lines elevates m⁶A_m level in mRNA which results in enhanced in vivo tumorigenicity and chemoresistance. Inhibition of the m⁶A_m methyltransferase of PCIF1/CAPAM partially reverses this phenotype. The authors showed that FTO/m⁶A_m axis constitutes a reversible pathway controlling CSC abilities that does not involve transcriptome remodeling, but rather modulates translation efficiency of selected m⁶A_m marked transcripts. This is an interesting work. But the manuscript need be improved to clarify the following questions.

1. It is somehow strange that the knockdown of FTO will not lead to the change of m⁶A in mRNA. The authors need to carefully measure the level of m⁶A in control and FTO knockdown samples.

We tried very hard to find any measurable change of m⁶A, both in mRNA (**Figure 4a**) and small RNA (**Figure S3d**), following FTO knockdown. Yet we did not see any significant variation. By opposition, any direct alteration of m⁶A effector expression (e.g. siRNA METTL14) results in changes of m⁶A level (**Figure S4a**). We concluded that FTO depletion did not affect significantly this modification in our cell model.

Altogether, this observation is not that surprising considering that: (1) targeting any other m⁶A effector did not affect sphere forming ability (**Figure 1b**); (2) m⁶A_m demethylation by FTO was previously reported [1]; (3) FTO effect varies from one cell line to another [2]; (4) in our system, demethylation activity takes place in the cytoplasm which emphasizes the importance of subcellular context (**Figure 5**) [3]; (4) PCIF1 KD in sh-FTO cells restores sphere forming potential to the level observed in control cells (**Figure 4g & S4e**).

2. In Figure 1 and 2, the authors found that FTO silencing promotes stem-like traits in colorectal cancer cell lines and confers resistance to chemotherapy by siRNA or sh-FTO knockdown. I would suggest the authors to perform parallel experiments by overexpressing FTO and then examine the effect upon FTO overexpression.

We did perform these experiments in CTC-derived cell lines, which exhibit basal lower FTO expression (with respect to primary and metastatic colorectal cancer cell lines) and enhanced stem-like traits [4]. As expected, overexpressing FTO inhibited sphere forming potential, chemoresistance and biomarker expression (**Figure3c-e**).

3. The authors had better provide evidences to demonstrate the purity of isolated mRNA for detecting the modifications of m⁶A_m, m⁶A, and Am since other RNA species also carry these modifications. Contamination of other RNA species in mRNA will cause the inaccurate quantification of tested modifications in mRNA, which may then lead to the wrong results.

We employed the same experimental pipeline than baseline studies in the field (e.g. [3]), with two rounds of polyA tail purification followed by rRNA removal. Yet, we now provide extensive evidence that demonstrate the purity of isolated mRNA samples as well as the robustness of our results:

- 1- Quantitative PCR demonstrate the efficiency of mRNA purification procedure (and depletion of rRNA) (**Figure S3b**)
- 2- We did not observe any change of m⁶A_m or m⁶A in purified small RNAs from the same cells (**Figure S3d**)
- 3- mRNA purification coincides with tremendous decrease of RNA modifications generally enriched in tRNAs (m¹A) and rRNA (m^{6,6}A, A_m) (**Figure 3a**).

4. As for the mass spectrometry measurement, the normal way to express the levels of these modifications is in the format of percentages of m6A/A, m6Am/A, Am/A. In this way, the authors can compare the measured levels of these modifications to previously reported levels of these modifications, which will also show the accuracy of the measurement. Right now the authors used the fold changes and the normalized data. With the current information, people don't know the real levels of these modifications. Moreover, I would suggest the authors to use m6Am/A, instead of m6Am/Am, to demonstrate the connect between m6Am and FTO.

As suggested, we have now modified the format of our results accordingly.

5. The author showed that low FTO expression in patient-derived cell lines elevates m6Am level in mRNA which results in enhanced in vivo tumorigenicity and chemoresistance. However, in the summary, the authors mentioned that "Even if limited to colon cancer, however, our findings point the way to deeper understanding of the CSC phenotype and have exciting implications for developing therapies that increase m6Am modification to cripple CSC-based metastases and resistance". These two are contradictory.

We apologize for this mistake and thanks the reviewer for pointing out this contradiction. We have fixed it now. It should read: "our findings point the way to deeper understanding of the CSC phenotype and have exciting implications for developing therapies that decrease m⁶A_m modification to cripple CSC-based metastases and resistance".

Reviewer #2 (Remarks to the Author):

I enjoyed reading this manuscript. The authors have performed quite comprehensive investigations of roles of RNA m6A regulators on colorectal cancer stem cells. They discovered that FTO is the only one that seems to affect CSC stem-like properties and resistance to chemo treatment. This part of the work is vigorous and well done. However, the main conclusion and the title are all incorrect or mis-leading. After reading this manuscript my conclusion was the role of FTO has less to do with mRNA modifications. The authors showed that the m6A level did not change much but cap m6Am level showed substantial change. These results are also misleading as they did not label correctly the exact percentage of m6Am/A and the change. Please do not show fold change in y-axis. A modification could change from 0.0001% to 0.0003% showing 3-fold change but has no functional relevance. If they label the exact level of m6Am/A the ratio is not going to be impressive.

We agree with this comment. We now have modified the format of our results accordingly. m⁶A_m/A ratio remains significant, except for **Figure 4b**, most certainly because we have only two CTC lines.

My biggest problem is that in the title and discussion the authors conclude that m6Am axis controls the cancer stem-like property of colorectal cancer. To justify they need to perform all functional characterization of PCIF1/CAPAM which installs the cap m6Am. This part is very thin and data not consistent at all. In fact PCIF1/CAPAM should have much stronger phenotype if the authors conclusion is correct. This is clearly not the case.

We apologize for the misleading title. We meant to say that FTO was acting through m6Am demethylation. We have modified it accordingly. Further, we now show that FTO-mediated m6Am demethylation occurs in the cytoplasm. By virtue of subcellular context [2], the cytoplasmic pool of FTO must certainly display some sort of substrate specificity (distinct from nuclear PCIF1). Nevertheless, PCIF1 silencing does “rescue” sphere forming ability in sh-FTO cells (**Figure 4g**), which confirms the involvement of m⁶A_m in our model.

The correct way to present the data is to analyze cellular localization of FTO first, measure nuclear versus cytoplasm m6A/A and m6Am/A levels (show exact percentage), suggesting that these changes may have minor effects on CSC as i) METTL3/14 KD did not affect CSC stem like properties; ii) PCIF1 KD could not recapitulate most CSC property changes observed with FTO KD. In this particular system FTO is likely working on a particular group of RNA m6A. This should be pursued in the future.

We are grateful to the reviewer for raising this issue. We now demonstrate that FTO is **both nuclear and cytoplasmic in our cell model (Figure 5a)**. Further, **m⁶A_m demethylation takes place in the cytoplasm (Figure 5c)**, not in the nucleus, in agreement with previous report [3]. Finally, change of FTO localization seems to stem from malignant transformation (**Figure 5d**). This was unexpected! Compartment-specific activity and substrate preferences would definitively explain why FTO and PCIF1 share partially overlapping mRNA targets [2]! Consequently, PCIF1 KD cannot recapitulate most CSC property changes.

We are currently following up on these observations to characterize the underlying molecular mechanism. There is a lot to be learned, especially why and how malignant transformation triggers FTO relocation to the cytoplasm.

Note that even the translation changes observed by the authors attributing to m6Am seem to be minor. I doubt the authors can use these translation differences to recapitulate phenotypes observed with FTO KD. It is perfectly fine to discover nice phenotypes and that none of the current pathways could explain the phenotypes suggesting new knowledge to be learned.

We did not expect strong changes at the translation level: First, they would have triggered significant transcriptome remodeling; second, our results are in line with PCIF1 KO cell that displays modest alteration at the translation level [5]. Nevertheless, we are now tempering our words: while we see differences at the translation level, we cannot exclude the involvement of other mechanisms. We intend to study more carefully the underlying mechanism in the future.

Note in other systems KD FTO reduces cell growth (HeLa, HEK, AML etc. etc). The authors need to be careful here. It seems to be dependent on targets of FTO.

We did not see any change of cell growth in colorectal cell lines (**Figure S1f**). We believe that FTO may target different RNA (as well as methylation sites, m⁶A vs m⁶A_m) depending on cell type. Sub-cellular context (FTO distribution, diversity in RNA species,...) may explain it.

Note that the overall survival of colorectal cancer is higher with lower FTO (check cancer database)! This is not consistent with shFTO promoting tumor initiation. The authors need to discuss this part.

The reviewer raises an excellent point. We did see this discrepancy between overall survival curve and **FTO mRNA expression**. However, our study clearly establishes that transcript-based analysis of clinical samples **must always been taken with a pinch of salt**: first, FTO expression is strongly regulated at the post-transcriptional level (**Figure 3a & b**); second, FTO displays compartment-specific methylation activity (**Figure 5c**); third, while maintaining constant level, FTO may re-localize to the cytoplasm throughout the course of tumorigenesis (**Figure 5d**). We now debate this part in the discussion of the manuscript.

In addition, see drug resistance in other systems: <https://www.nature.com/articles/s41422-018-0097-4>

Other questions:

Fig. 2f. It looks to me the protein levels of METTL3 and METTL14 both increased and ALKBH5 showed decrease.

We apologize for the lack of clarity. Indeed, chemo treatment may alter expression of m⁶A effectors. But their expression levels vary from one cell line to another (either up and down). Sole FTO is **always decreased**. We amended the text accordingly.

Fig. 4C is not consistent with Fig. 3C, note that m⁶A did not increase while FTO level reduced by 50%! This essentially suggested that m⁶A is not the target. And please show the absolute percentage. What is the level of PCIF1? This again indicates m⁶A is not the source.

This is an excellent point. The reviewer refers to m⁶A_m level in metastatic cell lines, which stays unaffected by a significant decrease of FTO level. Remarkably, metastatic cell lines show reduced PCIF1 level as well (by 50%, **Figure S4h**). We conclude that balanced decrease of writer AND eraser may explain unchanged m⁶A_m level (in comparison with primary cell lines).

The authors need to show KD PCIF1 in shFTO cells rescue phenotypes. Not a single experiment has been shown.

We performed this experiment. As expected, PCIF1 KD “rescues” sphere forming potential in sh-FTO cells to the same level observed in sh-CTL cells (**Figure 4g & S4e**). This implies a balance between FTO and PCIF1 enzymatic activities and supports the involvement of FTO in m⁶A_m demethylation.

I suggest the authors perform MeDIP on m⁶A and see if they identify any unique targets. The authors should analyze translation efficiency (protein level/mRNA) and seek correlation with m⁶A or m⁶A_m changes. To me it could be a specific group of RNA. Their Fig. 6a suggested m⁶A. The Wei ref is data from HEL cells. The authors should perform m⁶A MeDIP from their cells and preform analysis again.

In order to perform MeRIP-seq, we have initiated a collaboration with a lab that possesses this expertise (JY Roignant, CIG UNIL). Unfortunately, our collaborator had repeated issues with the polyclonal m⁶A antibody that they used (anti-m⁶A, SYSY No. 202003), which could arise from lot variability. Consequently, we could not get any enrichment of m⁶A-marked sequences following m⁶A immunoprecipitation (**cf metagene profile below**). Nevertheless, MeRIP-seq is known to be quite noisy (doi: <https://doi.org/10.1101/657130>) and suffers from poor resolution (about 150nt): it will not permit to distinguish m⁶A from m⁶A_m. Considering our latest results that strengthens the involvement

of m⁶A_m in FTO-dependent phenotype, we do not believe that MeRIP-seq analysis will provide major new insights into the molecular mechanism.

Lastly, if PCIF1 is involved can authors search cancer database and show its level correlation with survival and compare with FTO? I bet PCIF1 has no role in colon cancer.

Based on our experimental results, PCIF1 level may participate –at least partially- in the phenotype. We searched for it in cancer database and analyzed its level correlation with survival (see below). Indeed, PCIF1 does not seem to have a role in colon cancer survival. However, once again, PCIF1 KD cannot recapitulate most CSC property changes observed following FTO KD.

Kaplan plot for PCIF1 in COAD

Reviewer #3 (Remarks to the Author):

In this study, Relier et al investigated the role of mRNA m6Am modification on colorectal cancer stem cell (CSC) function. They focused on FTO, which has RNA demethylase activity and show that knockdown or overexpression of FTO alters the ability of CRC to grow as spheres in 3D and response to chemotherapeutic drug combinations, in vitro and in mouse xenografts. Overall, although the phenotypic data looks good and convincing for the most part, the role of regulation of mRNA translation by FTO is a major concern. In addition, the conclusions are overstated and not supported by the data. Specific comments/concerns are:

1. It is concerning that in the transcriptome analysis from shCTL vs shFTO cells, FTO expression does not change significantly (Table S4). This means that the knockdown of FTO was not sufficient and the conclusion on line 183 that FTO mainly acts at the level of translation may not be true. In fact, the authors may be missing out important targets of FTO that could help explain the observed phenotypes. This is a major concern. If the authors choose to address this, they should repeat the knockdown experiment and perform total RNA-seq. This should be done in two CRC lines and the genes that are commonly up- or down need to be selected for further analysis to try to see if they can explain the observed phenotype.

We apologize for this lack of clarity. FTO has an adjusted p-value 0.059 (with the stringent Benjamini-Hochberg correction procedure [7]), and obtains an (unadjusted) p-value of 2.6×10^{-5} . The choice of a 0.05 threshold for the adjusted p-value is frequent, but arbitrary. Moreover, p-value computation and adjustment procedures are approximations [6][7]. This p-value can be considered significant with a threshold of 0.06. The analysis of all RNA transcripts involved substantial background variation (which justify the p-value correction method). On the contrary, a qPCR test, which targets a single mRNA, is less prone to background variation and is more precise. Moreover, FTO is significantly decreased by quantitative PCR (see below) and immunoblot (which is the most important readout, Figure S1d). These results and the phenotype agree, which suggests the efficiency of FTO knockdown. The main goal of our study was not to suppress FTO expression, but to slightly decrease it to the level observed in CTC (about 50% inhibition, Figure 3a), which displays inherent CSC abilities [4]. We clearly demonstrate that 50-60% decrease is sufficient to increase significantly CSC properties (Figure 1g, 2c).

This finding strengthens the importance of our discovery: minor alteration of FTO expression and/or localization may have significant impact on cancer cell phenotype. Moreover, 90% silencing (e.g. using FTO siRNA, **Figure S1b**) did not further affect sphere forming ability (**Figure 1d vs S1e**).

n = 3 ; mean +/- S.E.M

2. The polysome data presented in the current manuscript does not explain the observed phenotypes. It could be that key genes are missed out due to incomplete knockdown of FPO. If shRNAs don't work very efficiently, the authors can knockout FTO using CRISPR/Cas9. HCT116 has been known as an excellent cell line for genome editing. The polysome experiments will have to be repeated to first make sure that the knockdown of FTO was substantial, followed by analysis of mRNAs whose translation is deregulated. One would expect that this approach may be useful in identifying the genes regulated by FTO that are important for its biological function.

As detailed above, FTO level decrease in sh-FTO cell lines is sufficient to promote CSC phenotype. Yet, we are now moderating our words in the discussion section: "while we see modest but significant differences at the translation level, we cannot exclude the involvement of other mechanisms."

3. In the sphere forming data in Figure 1C and 1D, for example, there error bars in the si-CTL are missing. Without error bars in the si-CTL, how did the authors calculate the p-value? This is a problem in majority of the Figures where the data for the control is set to 1. There are no error bars in the control.

We thank the reviewer for pointing this out. P-value was calculated correctly but the error bars were missing in the figures. We have now modified our figures accordingly.

4. Figure 2C: It will be good to show the tumor volume for the last time point for the sh-CTL and sh-FTO.

The tumor volume for the last time point is now indicated (**Figure 2c**).

5. The authors show that chemoresistance cells have 50% less FTO protein. Is this due to less FTO mRNA levels? This can be determined by qRT-PCR.

As suggested, we performed qRT-PCR. We found similar changes on mRNA levels in CRC1 cells but not in SW620. Therefore, change in protein abundance may be due to change on mRNA level in CRC1 but not in SW620 cells (**Figure S2e**). This confirms the importance of post-transcriptional control of FTO expression.

6. Line 123: What was the sub-lethal dose used in this experiment?

The sub-lethal dose was 0.2 X FIRI = 10 μ M 5-FU + 0.1 μ M SN38. It is now indicated both in the figure legend (Figure 2d) as well as in the main text.

7. Figure 1f: How could the authors calculate p-values without errors bars in the control sample (DMSO).

P-values were calculated in the correct way but the error bars on the figures were missing. Errors bars are now indicated. Calculation method is also detailed in Figure legend.

Looks like METTL3 levels were substantially reduced upon FIRI treatment and ALKBH5 went up.

We apologize for the lack of clarity. Indeed, chemo treatment may alter expression of m⁶A effectors **but not always**: e.g. ALKBH5 expression is slightly upregulated in CRC1 but does not change in SW620 following FIRI treatment. Sole FTO expression is **constantly decreased**. We amended the text accordingly.

8. The authors show that FTO expression is linked to chemoresistance. FTO kd results in increased tumor growth in response to CRC chemotherapy in vivo (Figure 2). This suggests that FTO inhibits tumor growth in response to chemotherapy. Would you not expect FTO levels to increase in response to chemotherapy, in contrast to what the authors show where they found that when CRC1 cells are treated with sub-lethal dose of chemotherapy, FTO levels decrease.

FTO expression (either high or low) does not affect cell growth in our cell lines, both in vitro (**Figure S1f**) and in vivo (**Figure 1g**). The whole point of this study is to demonstrate that decreased FTO expression coincides with enhanced CSC properties such as chemoresistance. **So far, every single experiment presented in this study support this conclusion**: (1) decreased FTO level bestows inherent resistance to FIRI; (2) Conversely, acquisition of chemoresistance concurs with decreased FTO expression.

FTO does not enhance tumor growth, but rather improve cell resistance to chemo treatment. Consequently, sh-FTO cells keep proliferating throughout FIRI treatment (**Figure 2c**), by contrast with control cells.

Also, what was the dose used for Figure 2f?

The dose for Figure 2f is 0.2 X FIRI (10 μ M 5-FU + 0.1 μ M SN38). This is now indicated both in the text and figure legend.

Can the authors perform PI staining and FACS analysis at this dose so that the effect on cell cycle and sub-G1 population is clear?

As suggested, we now provide PI staining and FACS analysis that show that treatment with 0.2 X FIRI triggers cell cycle arrest (G2/M) and significant increase of sub-G1 population (**Figure S2b**).

9. Figure 3d: It is unclear how the FTO protein levels in the Primary samples is not 1? How was this data normalized?

This data is normalized with respect to CRC1 cells.

10. The data in Figure 3 can be presented in a better way. In the text, the authors start with Figure 3c and 3d. Why not make these panels Figure 3a and 3b?

We modified the figure accordingly.

11. The conclusion in for the data in Figure 4 is overstated. The authors show that modulating FTO levels, alters m6Am/m6A ratio but these results are from a single cell line, CRC1. One would expect that this cell line has a high proportion of CSCs, but it still is a cell line, not a population of CRC1 cells enriched for CSCs.

We kindly disagree with this comment. First, the CTC lines display the highest proportion of CSCs [4], and the lowest FTO expression (**Figure 3a**). Second, FTO silencing experiments have been performed in two cell lines (CRC1 and SW620) and gave the same results (**Figure 4a**). Finally, FTO overexpressing experiments have been performed in two distinct CTC lines (**Figure 4c**). Altogether, the conclusion of this figure is based on solid data obtained from 4 distinct cell lines.

12. Line 105: change $p=0,00035$ to $p=0.00035$

13. Line 117: To evaluate “the effect” of FTO....

14. Line 115, change “inhibition’ to “knockdown” or “silencing”. Inhibition is used for inhibitors, not shRNAs.

Points 12 to 14 have now been fixed.

15.Line 146: Please specify in the text which RNase was used.

It is now specified in the text.

REFERENCES

1. Mauer, J., et al., *Reversible methylation of m(6)Am in the 5' cap controls mRNA stability*. Nature, 2017. **541**(7637): p. 371-375.
2. Shi, H., J. Wei, and C. He, *Where, When, and How: Context-Dependent Functions of RNA Methylation Writers, Readers, and Erasers*. Mol Cell, 2019. **74**(4): p. 640-650.
3. Wei, J., et al., *Differential m(6)A, m(6)Am, and m(1)A Demethylation Mediated by FTO in the Cell Nucleus and Cytoplasm*. Mol Cell, 2018. **71**(6): p. 973-985 e5.
4. Grillet, F., et al., *Circulating tumour cells from patients with colorectal cancer have cancer stem cell hallmarks in ex vivo culture*. Gut, 2017. **66**(10): p. 1802-1810.
5. Akichika, S., et al., *Cap-specific terminal N (6)-methylation of RNA by an RNA polymerase II-associated methyltransferase*. Science, 2019. **363**(6423).
6. Love, M.I., W. Huber, and S. Anders, *Moderated estimation of fold change and dispersion for RNA-seq data with DESeq2*. Genome Biol, 2014. **15**(12): p. 550.
7. Yoav Benjamini, Y. and Hochberg, Y., *Controlling the False Discovery Rate: A Practical and Powerful Approach to Multiple Testing*. JSTOR, 1995. **57**(1): p. 289.

Reviewers' comments:

Reviewer #1 (Remarks to the Author):

The authors addressed my questions. I don't have further comments.

Reviewer #2 (Remarks to the Author):

The results from the authors and their search of potential correlation of PCIF1/CAPAM in colorectal cancer confirmed my speculation that cap m6Am plays minimum role in the biological process the authors observed with FTO. Studies from other laboratories also showed minimum effects or phenotypes of PCIF1/CAPAM in affecting biological functions, confirming cap m6Am demethylation could not explain biological functions mediated through FTO. FTO works on other RNA substrates that bear polyA tail and are m6A methylated. I just cannot support publication of this manuscript.

Reviewer #3 (Remarks to the Author):

The authors have addressed my concerns.

We are grateful to the reviewers for their insightful and helpful comments to improve significantly our manuscript NCOMMS-19-379072 entitled "**FTO-mediated cytoplasmic m⁶A_m demethylation adjusts stem-like properties in colorectal cancer cell**".

Please, find below in blue text our point-by-point response to the reviews.

Reviewers' comments:

Reviewer #1 (Remarks to the Author):

The authors addressed my questions. I don't have further comments.

Reviewer #2 (Remarks to the Author):

The results from the authors and their search of potential correlation of PCIF1/CAPAM in colorectal cancer confirmed my speculation that cap m6Am plays minimum role in the biological process the authors observed with FTO. Studies from other laboratories also showed minimum effects or phenotypes of PCIF1/CAPAM in affecting biological functions, confirming cap m6Am demethylation could not explain biological functions mediated through FTO. FTO works on other RNA substrates that bear polyA tail and are m6A methylated. I just cannot support publication of this manuscript.

We respectfully disagree with this reviewer. He/she refers to other studies that showed minimum effects or phenotypes of PCIF1/CAPAM in affecting biological functions. Yet, no comparison can be made:

- (1) None of the other studies was performed in colorectal cancer cell lines while we tested 6 cell lines, including low passaged cell lines from patient samples. By contrast, two of the recent studies on PCIF1 use HEK cells (Boulias K. et al., Molecular Cell 2019; Akichika et al., Science 2019) where FTO is strictly nuclear (cf Nature communication paper from Koh et al, 2019)
- (2) We show below that FTO knockdown does not have the same effect on a breast cancer cell line (MCF7) while ALKBH5 knockdown decreases sphere forming ability (as previously reported in Zhang, C., et al., Proc Natl Acad Sci U S A, 2016. 113(14): p. E2047-56). Yet, ALKBH5 has not effect on sphere-forming ability in colorectal cell lines (**Figure 1c**). This demonstrates some sort of tissue specificity.

Impact of FTO or ALKBH5 inhibition (siRNA) on sphere forming ability. While FTO silencing has no effect on sphere forming ability, ALKBH5 silencing decreases stem-like properties as previously reported. Statistical analysis: Mean ± SD; *p < 0.05, unpaired t test.

- (3) FTO does not have the same targets than PCIF1. Indeed, PCIF1/CAPAM level did not affect CTC resistance to FIRI treatment (Fig S5a). Likewise, sub-lethal doses of FIRI, which promote CSC phenotype (Fig 2d), did not modify PCIF1 expression (Fig S5b). Nevertheless, these cells displayed reduced FTO expression (Figure S5b) as well as increased m⁶A_m level (Figure S5c). This suggests that **FTO and PCIF1 do not exhibit mere antagonistic activities but rather share a partially overlapping**

enzyme substrate specificity. We believe that distinct cellular distribution of these two effectors may explain such functional difference: PCIF1 methylates nascent RNA in the nucleus, FTO provides specificity in the cytoplasm!

We now provide clear evidence that m⁶A_m is central in the biological process we observed with FTO: **PCIF1/CAPAM knockdown in sh-FTO cells fully restores cell phenotype:** both sphere forming potential (**Figure 4f, S5f**) and chemoresistance (**Figure 4g**). Further, in order to ascertain whether FTO does not affect specific internal m⁶A sites, we employed methylated RNA immunoprecipitation sequencing (MeRIP-seq) from sh-CTL and sh-FTO cells. MeRIP-seq data analysis did not reveal any significant change between the two cell lines (**Fig. S5 and Table S7**).

Therefore, while we cannot exclude subtle alteration of internal m⁶A marks that would have been missed out due to the limit of detection of MeRIP-seq, our data clearly establishes cap-m⁶A_m residues of mRNA as the main substrate of cytoplasmic FTO in colorectal cancer cells.

Reviewer #3 (Remarks to the Author):

The authors have addressed my concerns.

REVIEWER COMMENTS

Reviewer #4 (Remarks to the Author):

I have read the Relier et al manuscript with interest. Even though cancer-related research is not my expertise, I saw a convincing characterization of a phenotype in colorectal cancer cells upon FTO depletion. As we are joining this review in a late stage after a comprehensive round of review and rebuttals, I will first focus on the key question of whether or not the manuscript provides conclusive evidence that the phenotype is mediated by m6Am versus m6A. In my view (yet, with the caveats listed below) the data shown by this manuscript is strongly suggestive that the phenotype is mediated by the former, rather than by the latter, as stated by the authors. The authors demonstrate this consistently using MS-based analysis, where they fail to find an effect on m6a levels yet find effects for m6Am. Of additional interest is the characterization that it's not the absolute change in protein level, but the change of subcellular localization of FTO that results in the change of total m6Am levels. Furthermore, in light of the relative quantifications now provided by the authors, it is clear (again, with the caveat below) that the m6Am levels are not in the range of 0.0001%, but substantially higher, and hence there is no reason to assume that the reported results lack functional relevance. Also the phenocopy and -rescue upon PCIF1 supports, that m6Am (or demethylation of m6Am plays a functional role).

Nonetheless, in reading this manuscript we did encounter several aspects raising questions/concerns which in our view would need to be addressed prior to publication:

Fig. 4b: Based on the MS quantifications, m6Am is present at a level of ~0.1-0.15 in CTC44 and CTC45 cell lines. Based on Fig4C, in CTC44 (oe-CTL) levels are roughly 4-fold higher (0.4%), whereas CTC45 is roughly 5 fold lower (0.03%). Is there any rationale for these dramatic inconsistencies? Can these measurements be trusted?

The same for Figure 5c. What explains the huge discrepancies in cytoplasmatic m6Am concentrations? Across replicates >10-fold differences are observed.

Fig. 4f-g: to evaluate the double knockdown of both FTO and PCIF1, it would be important to establish the effect of the individual knockdown of ONLY PCIF1 in the CRC1 cell line (without knockdown of FTO).

The authors refer to a meRIP analysis. Yet in their rebuttal, the authors note that this analysis did not have appeared to work, technically. In addition, none of the QC analyses presented in Figure S5 suggest otherwise. Basic QC features of m6A-meRIP analyses include an ability to de-novo identify the m6A motif, and the relative distribution of peaks along with transcripts that tend to be enriched near stop codons/terminal exons.

Despite the non-existing statistical significance for the m6Am level changes in primary, CTC and metastatic CRC cell lines (Fig. 4b) it is highlighted that the trend correlates with changes in FTO (Fig. 3a), yet the m6Am level correlated better with the detected PCIF1 transcript level in Fig S4g (panel title: "PCIF1 mRNA level does not change along with tumor progression.", despite showing a stronger trend than Fig. 4b).

In general, I felt that the mechanistic aspects of this work (primarily in Figures 6 and 7) were far from compelling. Figure 6 does not appear to be fixed for a p-value < 0.05, nor corrected for multiple testing. Figure 7c is a potentially misleading comparison because the comparison is performed across different genes. A far more compelling analysis would have been to compare the distribution of differences in translational efficiency between the same genes in WT versus PCIF1 depleted cells. In general, the literature about the roles of m6am is already quite contaminated, and we feel these two analyses do not add clarity to the field; in our view, they would best be eliminated.

We are grateful to the reviewer for his insightful and helpful comments to improve significantly our manuscript NCOMMS-19-379072 entitled "**FTO-mediated cytoplasmic m⁶A_m demethylation adjusts stem-like properties in colorectal cancer cell**".

Please, find below in blue text our point-by-point response to the reviews.

Reviewer comments:

Reviewer #4 (Remarks to the Author):

I have read the Relier et al manuscript with interest. Even though cancer-related research is not my expertise, I saw a convincing characterization of a phenotype in colorectal cancer cells upon FTO depletion. As we are joining this review in a late stage after a comprehensive round of review and rebuttals, I will first focus on the key question of whether or not the manuscript provides conclusive evidence that the phenotype is mediated by m⁶A_m versus m⁶A. In my view (yet, with the caveats listed below) the data shown by this manuscript is strongly suggestive that the phenotype is mediated by the former, rather than by the latter, as stated by the authors. The authors demonstrate this consistently using MS-based analysis, where they fail to find an effect on m⁶A levels yet find effects for m⁶A_m. Of additional interest is the characterization that it's not the absolute change in protein level, but the change of subcellular localization of FTO that results in the change of total m⁶A_m levels. Furthermore, in light of the relative quantifications now provided by the authors, it is clear (again, with the caveat below) that the m⁶A_m levels are not in the range of 0.0001%, but substantially higher, and hence there is no reason to assume that the reported results lack functional relevance. Also the phenocopy and -rescue upon PCIF1 supports, that m⁶A_m (or demethylation of m⁶A_m plays a functional role).

Nonetheless, in reading this manuscript we did encounter several aspects raising questions/concerns which in our view would need to be addressed prior to publication:

1/ **Fig. 4b**: Based on the MS quantifications, m⁶A_m is present at a level of ~0.1-0.15 in CTC44 and CTC45 cell lines. Based on **Fig4C**, in CTC44 (oe-CTL) levels are roughly 4-fold higher (0.4%), whereas CTC45 is roughly 5 fold lower (0.03%). Is there any rationale for these dramatic inconsistencies? Can these measurements be trusted?

The LC-MS systems can exhibit some variability concerning the analytical performances (MS intensity, retention time) over the time (months). Hence, the comparison of raw MS signals between experiments could be affected by these variations. As described by numerous authors, we use internal standards to normalize these variations. Thanks to these internal standards, results showed acceptable normalized intensities (<20%).

Cumulative factors may explain variations of m⁶A_m/A:

- (1) **Variations between experiments separated by several months** (up to 20%)
- (2) **The use of adenosine for relative quantification of m⁶A_m**. We observed significant fluctuations of adenosine from one experiment to another, most probably because of the nature of the sample, enriched in poly-A tails (whose length may vary). Adenosine signal from purified mRNA samples is also very strong, so strong that it can saturate the detector sometimes, forcing us to re-inject a smaller amount of sample. We do not encounter such difficulties with other nucleosides (U, C or G). Based on this observation, we do not believe that using adenosine for relative quantification of m⁶A_m is the most appropriate approach. We are actually evaluating alternative normalization procedure for relative quantification, including the use of isotope-labeled internal standards.

- (3) **“Low detectability” of m6Am.** m6Am is lower expressed than m6A. In fact, m6A/m6Am ratio is about 10 to 20 fold in basal condition and remains quite steady from one experiment to another. Detection of m6Am would not be an issue if we did not have to use “A” for relative quantification and inject smaller amounts of sample to avoid detector saturation.
- (4) **Cell transfection.** Figure 4b displays “wild-type” m6Am in wt cell lines. In Figure 4c, cells have been transfected with plasmids. This procedure may affect nucleosides level (both modified and unmodified).

Nevertheless, the “fold change” between experimental replicates remains quite steady (cf example below).

Thanks to this reviewer, we doubled-checked the raw data and **identified a loss of sensitivity in a series of experiments** shown in “Fig4c CTC45” and “FigS4e CTC45”. This loss of sensitivity affected molecules with “low detectability” such as m6Am (m6A was unaffected). We performed these experiments again and obtained an m6Am/A ratio much closer to the one shown in Figure 4b (see new figures “Fig4c CTC45” and “FigS4e CTC45”).

2/ The same for Figure 5c. What explains the huge discrepancies in cytoplasmic m6Am concentrations? Across replicates >10-fold differences are observed.

In this experiment, cytoplasm was separated from nucleus based on detergent-extraction procedure. This batch method introduces some degree of variability from one experiment to another, depending on the targeted protein(s) or RNA specie(s). Instead of repeating several times this experiment and pick the “most convenient” results, we chose to expose this experimental variability **while emphasizing the consistency of the observation**: cytoplasmic m6Am **is always** increased in sh-FTO cells in comparison with sh-CTL cells (while m6A remains unchanged).

3/ Fig. 4f-g: to evaluate the double knockdown of both FTO and PCIF1, it would be important to establish the effect of the individual knockdown of ONLY PCIF1 in the CRC1 cell line (without knockdown of FTO).

Excellent point. We now show that, unlike in CTC cell lines, PCIF1 knockdown in CRC1 does not affect basal CSC properties (sphere forming ability and chemoresistance) in CRC1 cells (Figure S4f, g). Yet, PCIF1 depletion in sh-FTO cells rescued the phenotype: sphere-forming ability (Figure 4f) as well as

chemosensitivity (**Figure 4g**). This emphasizes the fact that FTO and PCIF1 do not exhibit mere antagonistic activities but rather share partially overlapping enzyme substrate specificity.

4/ The authors refer to a meRIP analysis. Yet in their rebuttal, the authors note that this analysis did not have appeared to work, technically. In addition, none of the QC analyses presented in Figure S5 suggest otherwise. Basic QC features of m6A-meRIP analyses include an ability to de-novo identify the m6A motif, and the relative distribution of peaks along with transcripts that tend to be enriched near stop codons/terminal exons.

The Me-RIP worked technically but did not show any significant difference between sh-CTL and sh-FTO cells. We now provide adequate QC analyses:

(1) Identification of m6A motif. We applied the motif inference tool, Homer, on all peaks that are differentially enriched between IP and Input. We sought motifs of length 7, without mismatches; the entire sequence from the peak region is considered (we did not restrict the analysis to shorter windows centered around the peak maximum height). The most significant motif reported by Homer with a P-value of $1e-78$ is GAGGACT, which matches the DRACH motif on its last positions (GGACT). Moreover, the Homer motif in our condition is more specific than DRACH, as seen on the logo (**Table S7**) and on its average information content per position (1.82 bits, with a maximum of 2 bits per position); in addition, this motif matches 59% (2045) of target sequences. When Homer was run, with one mismatch allowed, for length 6 or 7, the same motif remains significant and appears as number 2 in the Homer list, showing that the motif is robust to variations of Homer parameters.

(2) Relative distribution of peaks along with transcripts (metagene profile). As expected, m6A is enriched near stop codons (**Figure S5d**).

5/ Despite the non-existing statistical significance for the m6Am level changes in primary, CTC and metastatic CRC cell lines (Fig. 4b) it is highlighted that the trend correlates with changes in FTO (Fig. 3a), yet the m6Am level correlated better with the detected PCIF1 transcript level in Fig S4g (panel title: "PCIF1 mRNA level does not change along with tumor progression.", despite showing a stronger trend than Fig. 4b).

We believe that considering protein level is more relevant than mRNA. Along this trend, FTO mRNA level does not correlate with FTO protein expression (**Fig. 3a and 3b**). Overall, this study shows that FTO level decreases and m6Am increases along with the acquisition of stem-like properties (e.g. chemoresistance). CTC lines display enhanced stem-like properties (cf Grillet et al, Gut 2017 doi: 10.1136/gutjnl-2016-311447) and high m6Am level (yet not statistically significant) in comparison with other cell lines (**Figure 4b**). PCIF1 protein level is equivalent between primary and CTC lines (**Figure S4j**). However, FTO protein expression is significantly decreased in CTC lines (**Fig. 3a**), which could explain the increased m6Am level. In metastatic cell lines, both FTO and PCIF1 proteins are decreased, which would explain why m6Am level is very similar to primary cell lines. Altogether, the ratio PCIF1/FTO (**Figure S4k**) correlates the best with m6Am levels observed in **Fig. 4b**.

6/ In general, I felt that the mechanistic aspects of this work (primarily in Figures 6 and 7) were far from compelling. Figure 6 does not appear to be fixed for a p-value < 0.05 , nor corrected for multiple testing. Figure 7c is a potentially misleading comparison because the comparison is performed across different genes. A far more compelling analysis would have been to compare the distribution of differences in translational efficiency between the same genes in WT versus PCIF1 depleted cells. In

general, the literature about the roles of m6am is already quite contaminated, and we feel these two analyses do not add clarity to the field; in our view, they would best be eliminated.

As emphasized in the manuscript, PCIF1 and FTO are not mere antagonists. We believe that cytoplasmic FTO targets a sub-pool of cap-m6Am mRNAs. Therefore, comparing the distribution of differences in translational efficiency between the same genes in WT versus PCIF1 depleted cells may not be an appropriate approach.

We certainly do not want to mislead people in the field. Consequently, Figures 6 and 7 will not appear in the main figures. That being said, it remains important to show that m6Am level does not impact cell's transcriptome and only very slightly translatoe in this cell model. Our data is robust and we did not expect so minor changes given the significant alteration of cell phenotype. While we cannot provide a solid conclusion regarding the mechanistic aspects, **we believe this data illustrates the actual shortfalls regarding m6Am role in the field.** This shall be discussed and made available to the scientific community.

These data are now part of supplementary Figure (**Figure S7**) and are only discussed in the last section of the paper.

REVIEWERS' COMMENTS

Reviewer #4 (Remarks to the Author):

I appreciate the effort of the response to my revision of the Relier et al. manuscript regarding the role of the m6Am-demethylase FTO in stem-cell-like properties of colorectal cancer cells. I was involved in this revision as a fourth reviewer to give my judgment regarding the involvement of m6Am or m6A, as we wrote before the data definitely shows the involvement of m6Am rather than m6A. I raised concerns regarding some issues in this manuscript and I see that the authors addressed each of them.

We do suggest that the authors explicitly address the dramatic differences in terms of the absolute difference in the mass-spectrometry measurements, and the potential underlying reasons, to prevent confusion by readers.

The m6Aseq quality control looks more reasonable now and all the other points were sufficiently addressed or explained from my perspective.

We are grateful to the reviewer #4 for his helpful comment to improve significantly our manuscript NCOMMS-19-379072 entitled "**FTO-mediated cytoplasmic m⁶A_m demethylation adjusts stem-like properties in colorectal cancer cell**".

Please, find below in blue text our point-by-point response to the reviews.

Reviewer comments:

Reviewer #4 (Remarks to the Author):

I appreciate the effort of the response to my revision of the Relier et al. manuscript regarding the role of the m6Am-demethylase FTO in stem-cell-like properties of colorectal cancer cells. I was involved in this revision as a fourth reviewer to give my judgment regarding the involvement of m6Am or m6A, as we wrote before the data definitely shows the involvement of m6Am rather than m6A. I raised concerns regarding some issues in this manuscript and I see that the authors addressed each of them.

We do suggest that the authors explicitly address the dramatic differences in terms of the absolute difference in the mass-spectrometry measurements, and the potential underlying reasons, to prevent confusion by readers.

The m6Aseq quality control looks more reasonable now and all the other points were sufficiently addressed or explained from my perspective.

We now address the differences in term of variation of LC-MS/MS measurements and potential underlying reasons in the manuscript.

Line 163: "Noteworthy, we sometimes observed significant fluctuations of adenosine measurements, most probably because of the nature of the sample, enriched in poly-A tails. Further, adenosine signal from purified mRNA samples was very strong and could saturate the detector. We do not encounter this issue with other nucleosides (U, C or G)."

Line 218: "While this batch method introduces some degree of variability from one experiment to another in terms of the absolute difference in the mass-spectrometry measurements, the result was clearly consistent across biological replicates."